# Dendrimers and Derivatives as Multifunctional Nanotherapeutics for Alzheimer’s Disease

**DOI:** 10.3390/pharmaceutics15041054

**Published:** 2023-03-24

**Authors:** Débora A. Moreira, Sofia D. Santos, Victoria Leiro, Ana P. Pêgo

**Affiliations:** 1i3S—Instituto de Investigação e Inovação em Saúde, Universidade do Porto, Rua Alfredo Allen 208, 4200-135 Porto, Portugal; 2INEB—Instituto de Engenharia Biomédica, Universidade do Porto, Rua Alfredo Allen 208, 4200-135 Porto, Portugal; 3FEUP—Faculdade de Engenharia, Universidade do Porto, Rua Dr. Roberto Frias, 4200-465 Porto, Portugal; 4ICBAS—Instituto de Ciências Biomédicas Abel Salazar, Universidade do Porto, Rua Jorge de Viterbo Ferreira 228, 4050-313 Porto, Portugal

**Keywords:** dendrimer, Alzheimer’s disease, nanomedicine, amyloid β, tau peptide, acetylcholinesterase, inflammation, oxidative stress, drug delivery

## Abstract

Alzheimer’s disease (AD) is the most prevalent form of dementia. It affects more than 30 million people worldwide and costs over US$ 1.3 trillion annually. AD is characterized by the brain accumulation of amyloid β peptide in fibrillar structures and the accumulation of hyperphosphorylated tau aggregates in neurons, both leading to toxicity and neuronal death. At present, there are only seven drugs approved for the treatment of AD, of which only two can slow down cognitive decline. Moreover, their use is only recommended for the early stages of AD, meaning that the major portion of AD patients still have no disease-modifying treatment options. Therefore, there is an urgent need to develop efficient therapies for AD. In this context, nanobiomaterials, and dendrimers in particular, offer the possibility of developing multifunctional and multitargeted therapies. Due to their intrinsic characteristics, dendrimers are first-in-class macromolecules for drug delivery. They have a globular, well-defined, and hyperbranched structure, controllable nanosize and multivalency, which allows them to act as efficient and versatile nanocarriers of different therapeutic molecules. In addition, different types of dendrimers display antioxidant, anti-inflammatory, anti-bacterial, anti-viral, anti-prion, and most importantly for the AD field, anti-amyloidogenic properties. Therefore, dendrimers can not only be excellent nanocarriers, but also be used as drugs per se. Here, the outstanding properties of dendrimers and derivatives that make them excellent AD nanotherapeutics are reviewed and critically discussed. The biological properties of several dendritic structures (dendrimers, derivatives, and dendrimer-like polymers) that enable them to be used as drugs for AD treatment will be pointed out and the chemical and structural characteristics behind those properties will be analysed. The reported use of these nanomaterials as nanocarriers in AD preclinical research is also presented. Finally, future perspectives and challenges that need to be overcome to make their use in the clinic a reality are discussed.

## 1. Introduction

Dementia describes a number of disorders that entail the loss of cognitive function and behavioural abilities in a way that significantly interferes with the person’s daily life and activities [1]. According to the World Health Organization (WHO), more than 55 million people worldwide are living with dementia, and this number is expected to triple by 2050 [2]. Within dementia disorders, Alzheimer’s disease (AD) is the most prevalent form, corresponding to 60–70% of all cases [3]. AD is characterized by progressive neuronal loss, which leads to a continuous and irreversible loss of memory and thinking skills, ultimately resulting in the loss of the ability to carry out the simplest tasks [4]. It is associated with the aggregation of amyloid β (Aβ) and hyperphosphorylated tau, but it has also been linked to cholinergic deficit, oxidative stress, mitochondria dysfunction, inflammation, and synaptic changes. Hence, AD poses as a molecularly complex disease. Up until recently, the available therapies could act only on symptoms’ attenuation. There are two main therapeutic approaches approved for symptom attenuation—cholinesterase inhibitors (donepezil, rivastigmine, galantamine) and N-methyl D-aspartate (NMDA) receptor antagonists (memantine). Both strategies attempt to reduce excitotoxicity, enabling synaptic communication and memory preservation, and avoiding further damage [5]. However, neither of these therapeutic approaches can modify or stop the progression of the disease, thus cognitive impairment is not only irreversible but also inevitable.

In June 2021 and January 2023, two new disease-modifying therapies have been approved for the treatment of AD—aducanumab (Aduhelm^®^) and lecanemab (Leqembi^®^), both commercialised by Biogen and Eisai [6,7]. These immunotherapies shed a light on the field. Nonetheless, their approval was wrapped in controversy as their safety is still an issue [8], and they offer a therapeutic solution only for the early stages of AD [6,7]. Hence, new therapeutics, namely disease-modifying therapeutics, are still needed.

AD drug development has a 99.6% failure [9]. The high rate of failure can be traced back to several reasons. First, drug delivery to the brain is still a challenge because of the blood-brain barrier (BBB), which prevents most pharmaceuticals from reaching the brain [10]. When pharmaceuticals have low BBB permeability, there is a need for high doses administration so that drug concentration in the brain reaches the therapeutic level, which can cause severe side effects. Therefore, new therapeutics must have good BBB permeability and, ideally, brain targetability. Second, as AD poses as a molecularly complex disease and its onset is still unclear, disease-modifying therapies are more likely to succeed through a multifunctional approach. Yet, multifunctional approaches have been very few [11].

Nanotechnology poses a novel and robust strategy that can overcome these issues. Due to their nanoscale size, structure and customizable surface, nanoparticles can cross the BBB, deliver several therapeutics at once in a controlled and specific way, and improve the pharmacokinetic and pharmacodynamics profile of these therapeutics [12]. In this way, nanoparticles can increase the therapeutics’ concentration at the target site and decrease unwanted toxicity in other sites [13]. Additionally, since they can carry different therapeutics at once, multifunctional approaches are achievable. Within nanoparticles, those based on dendrimers stand out for AD treatment.

Dendrimers are highly branched macromolecules with globular shape and a densely packed surface [14]. Due to their intrinsic characteristics, dendrimers are especially interesting for biomedical applications. Their globular shape, predictable molecular weight (MW), well-defined and customizable structure, low polydispersity, and high number of surface functional groups make them excellent drug delivery systems [15,16]. Dendrimers enable the carriage of different therapeutics in three ways—by encapsulation of drugs within the internal cavities of their structure, by covalent bonding to the functional groups or by non-covalent interactions formatting a dendritic nanoparticle [14]. Since the interior of the dendrimer can have an entirely different chemical environment than the periphery, encapsulation can be especially interesting for hydrophobic drugs with poor pharmacokinetic and pharmacodynamic properties. On the other hand, therapeutics covalently bonded to the dendrimers’ functional groups permit to deliver a high and controlled number of molecules at once while allowing a targeted and controlled delivery. In addition, the controlled and elevated number of functional groups at the dendrimers’ surface allows them not only to fine-tune their surface properties and consequently control how they interact with biomolecules and cellular components, but also allows them to attach different cargos and/or targeting moieties to the molecules. The functional terminal groups confer dendrimers a valuable characteristic—multivalency.

In addition to their excellent characteristics as nanocarriers, dendrimers have demonstrated their use as a drug per se [14,17]. Different architectures of dendrimers have shown antimicrobial, anti-viral, anti-inflammatory, anti-prion, and most importantly for the AD field, anti-amyloidogenic properties [14,17,18]. Several types of dendrimers have been described to interact with amyloid species preventing not only its aggregation but also its neuronal toxicity [19,20,21,22]. Additionally, they have been described to have an inhibitory effect on acetylcholinesterase activity [23,24,25], protect synapses, and improve memory [26]. By allying the intrinsic properties of dendrimers with a therapeutic load, a multivalent, multifunctional, and multitarget approach could be developed.

In this literature review, the properties of dendrimers and derivatives that make them excellent AD therapeutics per se and nanocarriers will be discussed. In the next sections, examples of numerous dendritic structures (dendrimers, derivatives, and dendrimer-like structures, such as hyperbranched polymers), which presented relevant properties for the treatment of AD will be presented and, whenever possible, conclusions on the chemical and structural characteristics that confer them those properties will be drawn.

## 2. Alzheimer’s Disease—Pathogenesis and Therapeutical Routes

Alzheimer’s disease was first described by Alois Alzheimer in 1906 [27]. Since then, a great deal of research has been done toward understanding the disease and its onset, yet its pathogenesis is still a debate. The molecular hallmarks of AD are the extracellular fibrillar aggregation of Aβ peptide in senile plaques and hyperphosphorylation of tau protein, which leads to its aggregation in intracellular neurofibrillary tangles (NFT) [28]. Furthermore, AD is associated with a cholinergic deficit, excessive reactive oxygen species (ROS), mitochondria dysfunction, inflammation and synaptic changes in the cerebral cortex, hippocampus, and other areas of the brain essential for cognitive and memory functions [29,30].

Despite the elevated knowledge of the molecular characteristics of the disease, there is not a clear onset. Based on the molecular features of AD, several hypotheses have been proposed for AD pathogenesis—amyloid cascade, tau, cholinergic, excitotoxicity, and mitochondrial cascade hypothesis. The most accepted hypothesis is the amyloid cascade hypothesis because it is supported by the genetics of early-onset familial AD (mutations in amyloid precursor protein (APP), presenilin 1 (PS1; also known as PSEN1) and PS2 (also known as PSEN2) genes), and late-onset AD (ε4 allele of the apolipoprotein E (APOE) gene) [28]. Transgenic mice expressing the mutations of familial-associated AD genes (APP/PS1/PS2) progressively developed brain Aβ plaques and memory deficits, reinforcing the relation between amyloid deposition and memory impairment [31]. Moreover, the presence of senile plaques is one of the earliest markers of AD, preceding the clinical symptoms by 15–20 years [31]. In this pathogenesis hypothesis, the disease onset poses as follows: the transmembrane protein APP is cleaved by β-secretase (also known as β-site APP cleaving enzyme 1; BACE1) and γ-secretase, producing various isoforms of Aβ peptide [28]. The increase of the isoform Aβ (1–42), which is the most prone form to aggregate, will result in the formation of amyloid fibrils and its intermediary species, translating into neurotoxicity and triggering the other disease molecular hallmarks [32].

Amyloid fibrils are highly ordered, β-sheet rich misfolded protein aggregates, often insoluble, that accumulate abnormally in tissues leading to toxicity [33]. Structurally, they are unbranched 2–20 nm diameter and several µm long structures, characterized by β-sheets motifs where individual β-stands are stacked with a perpendicular orientation to the fibril main axis [33]. These amyloid fibrils are formed in a nucleation-dependent way, following a sigmoid kinetics curve (Figure 1). Here, the aggregation process starts with the initial lag phase (nucleation), which is followed by a rapid elongation phase and saturation. In the nucleation phase, soluble monomers associate together producing oligomers, forming nuclei for further elongation and fibril formation. Since monomers’ self-association is thermodynamically unfavourable, this step is slow and is the limiting step of amyloid formation [34]. After nuclei formation, thermodynamically favourable elongation starts to yield protofibrils and filaments, which are converted to mature fibrils at the saturation step [34].

Mature Aβ fibrils tend to form extracellular senile plaques, which have been described to compromise synapse communication and cerebral integrity [35,36,37]. Nonetheless, the presence of Aβ plaques correlates poorly with the disease severity and cognitive impairment [38,39]. Several studies have suggested that the most pathogenic amyloid specie is oligomers because they can interact with cellular structures [40,41]. In fact, Aβ oligomers have been described as synaptotoxic and are thought to induce tau pathology, the downstream imbalance of ROS, inflammation, and ultimately cell death [28,29]. Oligomers are globular aggregates that lack a well-defined secondary structure [42]. Due to their non-specific secondary structure and small size (diameter range ~5–15 nm and MW ~20–1000 kDa [42]), they expose hydrophobic groups on the surface and easily interact with membrane lipids, membrane receptors, and cell organelles [40]. They have been described to increase the membrane fluidity, lead to the formation of ion channels in the lipid bilayers, and trigger a variety of downstream signalling pathways via membrane receptors that negatively impact cell function and survival. In addition, they have been described to interact with the mitochondria [43,44,45], endoplasmic reticulum [46,47], lysosomes [48], and nucleus [49,50]. Thus, the Aβ oligomers can severely compromise neuronal integrity and trigger other disease hallmarks.

Taking into consideration the described Aβ neurotoxicity, compounds that can inhibit, reverse, or reduce the aggregation of amyloids while avoiding oligomers’ toxicity could represent powerful approaches for AD therapy. The newly approved immunotherapies aducanumab and lecanemab were shown to interact with Aβ species and reduce Aβ burden, leading to a slowing in cognitive decline in early AD and mild cognitive impairment (MCI) patients [51,52,53,54]. These pharmaceuticals are human monoclonal antibodies that selectively target pathologic forms of Aβ, either oligomers/protofibrils (lecanemab) [55] or fibrils (aducanumab) [56]. Both antibodies showed to reduce the brain amyloid burden, yet the clinical relevance of this reduction was different. In a phase III clinical trial (EMERGE trial, NCT02484547), a monthly intravenous administration of aducanumab (10 mg/kg) led to the slight (yet significant) slower cognitive decline of treated patients, compared to the placebo groups [54]. However, in another phase III clinical trial (ENGAGE, NCT02477800), it failed to meet its primary outcome [57], challenging its clinical relevance. On the other hand, the lecanemab clinical effect leaves no room for doubt. Phase II and III clinical trials of lecanemab showed that a twice-monthly 10 mg/kg intravenous administration reduced brain amyloid and slowed cognitive decline by 27% over 18 months [51,52]. These trials demonstrated that targeting soluble protofibrils translated into much higher clinical effectiveness than targeting fibrils, demonstrating the relevance of oligomers and small Aβ aggregates. Moreover, they show that Aβ plays an important role in AD pathology and reinforce the amyloid cascade hypothesis. Nonetheless, both therapies are associated with severe side effects, such as cerebral edema, nausea, and confusion [52]. Since their safety is still an open issue, the approval of aducanumab and lecanemab for commercialization by the FDA is still controversial.

The second most important hallmark of AD is the hyperphosphorylation of tau protein and intracellular NFT. The tau protein is a member of the microtubule-associated proteins, and its main cellular function is to contribute to the elongation and stabilization of microtubules [58]. It is mainly expressed in neurons and at low levels in glial cells, and has been implicated in neuronal maturation, maintenance of cytoarchitecture, and synapses [59]. Even though tau misfunctions are not exclusive to AD, there is a strong correlation between cognitive dysfunction and memory loss with the NFT load and localization [38,60]. In addition, several knock-out animal models have shown that a loss of tau function is detrimental to cognition and memory [59,61]. Therefore, targeting tau malfunction could be critical to creating a disease-modifying therapy.

There are three main strategies to decrease tau toxicity: inhibition of tau aggregation, blockage of tau phosphorylation, or clearance of phosphorylated tau by immunotherapy [28,59]. The most widely pursued strategy is a reduction in tau hyperphosphorylation either by kinase inhibitors or phosphatase activators [59]. Three major classes of kinases phosphorylate tau [59]: proline-directed kinases (e.g., glycogen synthase kinase 3 beta (GSK-3β) [62] and cyclin-dependent kinase 5 (CDK5) [63]), non-proline-directed kinases (e.g., tau-tubulin kinases (TTBK) [64] and microtubule affinity regulated kinases (MARK) [65]), and tyrosine kinases [66]. Oppositely, tau dephosphorylation is mainly performed by protein phosphatase 2A (PP2A) [67]. Currently, there are five new drugs in clinical trials to reduce tau phosphorylation. Their target is to inhibit GSK3-β (Tideglusib, lithium, Valprotate) or tyrosine kinase (Nilotinib), or to activate PP2A (sodium selenate) [59].

Despite the recent success of aducanumab and lecanemab, all the other clinical trials that targeted a single pathological characteristic of AD have failed to show clinical benefit [9,68]. Therefore, targeting more than one molecular hallmark of the disease in a multifunctional approach is more likely to succeed and translate into positive clinical outcomes [58]. Besides Aβ and tau pathological hallmarks, AD has other molecular hallmarks associated, such as inflammation, ROS imbalance, mitochondria dysfunction, and calcium imbalance. Any strategy that could target Aβ toxicity and/or tau dysfunction and, at the same time, associate therapeutical characteristics for these hallmarks could represent a successful disease-modifying therapy.

## 3. Dendrimers—A Multivalent and Multifunctional Nanocarrier

Dendrimers were first described by Vögtle et al. in 1978 [69] as “cascade molecules”. Only later, after further work by Denkewalter, Tomalia, Newkome, Frechet, and co-workers, these highly branched molecules were termed dendrimers. Dendrimers get their name due to their characteristic organization. They consist of a bi- or multi-functional core molecule to which they are covalently linked to the branching building units [14]. These branching units are organized in generations, which correspond to the layers of branching points when going from the core towards the surface (Figure 2). At the surface, dendrimers present a large and controlled number of functional terminal groups that define their surface properties. The higher the generation of the dendrimer is, the more functional end groups it has, and the more densely packed its surface is [70]. This translates into three important aspects: (1) the core and interior become shielded from the surroundings, which allows to create a distinct chemical environment in the interior of the dendrimer; (2) due to the interior shielding, the characteristics of the dendrimer are mainly dominated by the polyvalent surface; and (3) it allows dendrimers to serve as “dendritic boxes” that can carry, inside their internal cavities, a small compound.

Dendrimers’ structure and MW are highly predictable due to their synthesis methodology, which is iterative and involves a series of repetitive growth and activation steps [14]. Classically, there are two approaches to synthesise dendrimers—the divergent and the convergent strategy (Figure 3). In the divergent strategy (Figure 3a), the dendrimer is grown generation by generation from the core to the periphery by the addition of new repeating units. In the convergent strategy (Figure 3b), the first part of the synthesis is to grow the dendrimer’s branch or dendron. Then, several dendrons are linked together by reacting with the multifunctional core, yielding a complete dendrimer [14]. In both synthetic approaches, these reactions include deprotection/activation steps of the branching points and growth of the dendritic structure to create a new generation (Figure 3). Regardless of the synthetic approach, the final number of surface functional groups/multivalency is predictable and controlled.

Comparing to other nanostructures, dendrimers have several advantages that give these macromolecules an edge in the biomedical field. First, due to their synthetic route, dendrimers have a well-defined structure and MW, making them defined chemical entities [71]. This feature is decisive for biomedical applications as it allows a better prediction of their function, their biological effect, and further contributes to performance reproducibility. Secondly, when optimized, the synthesis of dendrimers renders near monodisperse nanostructures. Other polymeric and lipidic nanoparticles are usually more polydisperse populations than dendrimers as the fabrication process is stochastic [72]. The low polydispersity of dendrimers and their well-defined structure are favourable characteristics that facilitate the regulatory process of approval for clinical use. Thirdly, the dendritic structure can be designed and modulated as required. By changing the synthesis of the dendrimer, its inner chemical structure and surface groups can be fine-tuned to possess the physicochemical properties one wishes. This characteristic confers dendrimers the ability to carry virtually any therapeutic or molecules, either by encapsulation or covalently bonded to the surface, as their hydrophilic/hydrophobic nature and surface moieties can be modulated. Fourthly, dendrimers are macromolecules with low nanometer size range (<15 nm) and globular shape, allowing them to cross several in vivo barriers, such as the BBB. Additionally, they resemble other biomolecules, such as proteins and enzymes, and can serve as mimetics for these biomolecules [70]. For example, generation 9 (G9) acetylated PAMAM dendrimers encapsulating platinum (Ac-G9/Pt) have been used to mimic catalase [73]. Lastly, the multivalency of dendrimers allows them to interact with biotargets in a polyvalent manner, permitting higher affinity and avidity. Multivalence is also a valuable characteristic for nanodelivery as it allows the binding of a high number of therapeutical moieties translating into an increased delivery. Their numerous surface groups also permit the multiple functionalization, which can not only contribute to the polyvalent interaction with biological targets but also confer dendrimers multifunctionality.

In the biomedical field, the most researched dendrimers are poly(amido amine) (PAMAM), poly(propylene imine) (PPI), phosphorus-based dendrimers, poly(L-lysine)-based (PLL), carbosilane, poly(ether)-copoly(ester)(PEPE), poly(ether imine) (PETIM), polyphenylene dendrimers, and gallic acid-triethylene glycol (GATG) dendrimers (Figure 4) [14]. PAMAM and PPI dendrimers were the first dendritic structures to be described [69,74] and later were made commercially available [16]. Consequently, they are the most widely used dendrimers. They both have polyamine groups on their surface, yet their synthesis and branching units are distinct. Comparing the two dendritic structures, PPI dendrimers are slightly smaller and have a more densely packed surface than PAMAM dendrimers with the same generation because of their shorter branching unit.

So far, different types of dendrimers have been successfully used as nanodelivery systems or as drugs per se. Examples of their use as nanocarriers are two PAMAM-based transfection agents for in vitro assays (SuperFect^®^ and PrioFect™, marketed by Qiagen (Düsseldorf, Germany) and Starpharma (Melbourne, Australia), respectively), and the PLL dendrimer holding gadolinium (III)-DOTA chelate groups at its surface (Gadomer^®^-17, invivoContrast) use as a pre-clinical research contrast agent [14]. Successful uses of the dendrimers as a drug per se include VivaGel^®^ and VIRALEZE™ from Starpharma. Both VivaGel^®^ and VIRALEZE™ have astodrimer sodium (SPL7013) dendrimers as active ingredients [75]. SPL7013 is a G4-PLL-based dendrimer, presenting negatively charged terminal groups, formulated to be used as an antiviral and antibacterial agent. VivaGel^®^ has been explored by Starpharma as a water-based mucoadhesive gel to be delivered vaginally, to treat and prevent bacterial vaginosis, and to serve as a protection against sexually transmitted infections (transmission of genital herpes (HSV-2), human immunodeficiency virus (HIV), and human papillomavirus (HPV)). It has been proven safe in female and male individuals, as well as clinically effective against bacterial vaginosis in several Phase II and Phase III clinical trials (for details see [76,77]). VivaGel^®^ is also available as a condom lubricant to provide extra protection against sexually transmitted infections [14]. VIRALEZE™ has been proposed as a virucidal and antiviral agent to prevent viral respiratory infections, such as the flu, coronavirus disease 2019 (COVID-19), and Respiratory Syncytial Virus Infection [78]. Formulated as a nasal spray, VIRALEZE™ has been shown to prevent SARS-CoV-2 genome copies in 99.9% and reduced the infectivity by >95%, both in vitro and in vivo [79,80]. In a Phase I Australian clinical trial (ACTRN12620001371987), it was proven safe and well-tolerated in humans [81]. These formulations are not systemically absorbed, and both are currently registered as medical devices in the UK, Europe, and Southeast Asia [75]. VivaGel^®^ is also available in South Africa, Australia, and New Zealand [75].

Besides VivaGel^®^ and VIRALENZE™ no other dendritic structure is commercially available for clinical applications. Nonetheless, several dendrimers are currently in clinical trials [77]. G4 hydroxyl-terminated PAMAM (PAMAM-OH) dendrimers, explored by Ashvattha Therapeutic (Redwood City, CA, USA), are being tested in clinical trials as drug delivery systems to inflammatory cells and tumours, as they have shown intrinsic targetability to these cells [82,83,84,85,86]. One example is OP-101, a G4 PAMAM-OH dendrimer functionalized with 24 moieties of the anti-inflammatory agent N-acetyl cysteine (NAC). OP-101 was proven safe for intravenous (NCT03500627) and subcutaneous administration (NCT04321980) in Phase I clinical trials. Moreover, it showed the ability to reduce the risk of death and the need for mechanical ventilation in severe COVID-19 infection in a Phase IIa trial (NCT04458298) [87]. By the same manner, a cationic peptide-based dendritic structure (KK-46) is currently in clinical trials (Phase I (NCT05184127) and Phase II (NCT05184127)) for the delivery of a siRNA for silencing SARS-CoV-2. This agent reduces SARS-CoV-2 infectability by inhibiting its replication [88].

G4-PAMAM-OH dendrimers are also under clinical testing as nanocarrier of therapeutics for age-related macular degeneration (NCT05105607; NCT05387837) [89] and as a contrast agent for positron emission tomography (PET) to patients with amyotrophic lateral sclerosis and other inflammation-associated disorders (NCT05395624) [90].

Another dendrimer family currently under clinical trials is the Dendrimer Enhanced Product (DEP^®^) Drug Delivery, developed by Starpharma. DEP^®^ is a G5 PEGylated PLL dendrimer explored as a drug delivery system for several drugs [91,92]. It is currently in Phase I/II clinical trials for the delivery of docetaxel (2016-000877-19), cabazitaxel (2017-003424-76), and irinotecan (2019-001318-40), in all of which are being used as anticancer agents for advanced solid tumours [77,93,94,95]. DEP^®^ has also been explored as a delivery system for the AstraZeneca’s cancer drug AZD4320, rendering the AZD0466 compound [91,92,96]. The safety, tolerability, and maximum tolerated dose of AZD0466 have been evaluated in a Phase I clinical trial in patients with advanced solid tumours, lymphoma, multiple myeloma, or hematologic malignancies (NCT04214093) [97]. It is currently in Phase I/II clinical trials as an anticancer agent for advanced haematological malignancies (NCT04865419) [98] and advanced non-Hodgkin lymphoma (NCT05205161) [99].

Besides antiviral and antibacterial properties, different types of dendrimers have been described to have other intrinsic properties, such as anti-inflammatory, antioxidant, chelator capacity, anti-prion, and anti-amyloidogenic properties [14,17,18]. As these properties could translate into a clinical improvement in the context of AD, dendrimers pose as a powerful tool for the treatment of AD. In the next sections, examples of dendrimers that presented such properties will be discussed and the characteristics that influence these properties will be dissected.

Other dendrimer-like structures, such as hyperbranched polymers (HBP), also possess interesting biomedical properties. HBP have a very branched structure with a high number of surface terminal groups conferring them multivalency. This feature grants them biological properties comparable with the dendrimers’ ones. Nonetheless, their structure and synthesis distinguish them from dendrimers. While dendrimers have a well-defined and near monodisperse 3D hyperbranched structure with well-established branching units, HBP possess dendritic units and linear units within their macromolecular framework, resulting in irregular structures [100]. Additionally, HBP are frequently synthesized by a single polymerization reaction, which results in low reproducibility and polydisperse structures and nanostructures. On the other hand, the iterative and multi-step synthesis process of dendrimers results in well-defined, near monodisperse, and highly reproducible structures [100].

Within HBP, the dendritic polyglycerol (dPG) family stand out for its biomedical applications. dPGs are based on a biocompatible polyether backbone which possesses a high branching degree, translating into a high number of surface functional groups. When functionalised with terminal sulphates (dPGS), they have been shown to possess anti-inflammatory and anti-viral properties on their own and the ability to carry therapeutics [101]. For this reason, this family of HBP will be included in this review as their chemical/structural characteristics can bring important insights into the relevant traits that enable the biological properties of dendritic structures.

## 4. Dendrimers as Anti-Amyloidogenic Agents

The anti-amyloidogenic properties of dendrimers were described by accident for Prion Protein (PrP). The PrP is a normal host protein that can attain an abnormal, fibrillar, and infectious conformation (denoted PrP^Sc^), grossly different from its normal and non-fibrillar conformation (PrP^C^). The PrP^Sc^ is associated with several pathologies, so-called prion diseases, that include Creutzfeldt-Jakob disease in man and bovine spongiform encephalopathy in cattle (mad cow disease) [70]. In 1999, Supattapone and his group transfected a murine neuroblastoma cell line (N2a) with DNA coding for an epitope-tagged PrP using the transfection reagent SuperFect™ (Qiagen) to create a persistently PrP^Sc^-infected murine neuroblastoma cell line (ScN2a). The results showed that, even though the transfection was successful and cells were expressing the epitope-tagged PrP, it did not result in the expression of PrP^Sc^. Transfection of the same gene using different transfection agents (non-dendritic) resulted in the expression of PrP^Sc^. Therefore, the transfecting reagent SuperFect™ was inhibiting the conversion of PrP^C^ to PrP^Sc^. Then, the group explored the anti-amyloidogenic and anti-prion properties of different generations of PAMAM and PPI in ScN2a cells and on PrP^Sc^-containing brain homogenates from prion-infected animals [102]. This study demonstrated for the first time that PAMAM and PPI could not only inhibit the fibrillation of PrP but also disintegrate previously formed PrP^Sc^, in a generation-dependent way.

Based on the structural homology of the segment 185–208 of human PrP and the peptide Aβ [103], the anti-amyloidogenic properties of G3 PAMAM dendrimers were studied in Aβ (1–28) and PrP (185–208) [104]. These dendrimers exhibited similar anti-amyloidogenic properties for both peptides and could inhibit fibrillation in a concentration-dependent manner. Hence, the anti-amyloidogenic properties of PAMAM dendrimers were not PrP-specific but could interfere with the fibrillation of other proteins. Since then, this property has been explored in different dendritic architectures and demonstrated in several fibrillar proteins. These properties were further described in PAMAM [19,104,105,106,107], PPI [22,108], cationic phosphorous dendrimers (CPD) [21,24,109], GATG dendrimers functionalised with morpholine groups [110] or gallic acid [111], carbosilane dendrimers [112,113], viologen-phosphorus dendrimers (VPD) [114,115,116,117], and different types of glycodendrimers [20,26,108,118,119,120,121,122,123]. Most of these dendritic macromolecules have demonstrated the ability to inhibit fibrillation, degrade pre-existent aggregates, and protect cells from the toxic effects of the aggregation species for amyloid species of Aβ, PrP, α-synuclein, and others. More importantly, glycodendrimers have shown to hinder the Aβ burden in vivo [26,118,123]. The chronical intranasal administration of G4 histidine and maltose shell PPI dendrimer (G4HisMal) resulted in a significant memory improvement in APP/PS1 mice, compared with control APP/PS1 mice treated with PBS [26]. G4HisMal reduced the formation of non-fibrillar oligomeric amyloid aggregates and the number of fibrils in vivo, which can explain the memory improvement [123]. G4HisMal administration also led to the preservation of synaptic markers like Psd95, synaptophysin, and drebrin, suggesting synapse protective properties as well [26].

Even though distinct dendrimers possess anti-amyloidogenic properties, their characteristics are different and can bring important insights into how the dendrimers interact with the amyloid species and affect the amyloidogenic process. Here, the dendrimer/peptide ratio, generation/size, and the nature of the dendritic surface can influence the anti-amyloidogenic properties of the particle. Table 1 summarizes the results of several studies on the anti-amyloidogenic properties of dendrimers. In the following paragraphs, the structural characteristics that influence these properties are described and discussed.

### 4.1. Dendrimer/Peptide Ratio

Most of the reports have shown that the inhibitory effect of dendrimers is dependent on the dendrimer/peptide ratio (Table 1). At a low dendrimer/peptide ratio, dendrimers were found to accelerate aggregation and increase the final fibril amount, whereas at a high dendrimer/peptide ratio they slowed down aggregation and decreased the number of fibrils. This behaviour is typical of inhibitors that can break fibrils [19]. When at low concentration, they speed up the aggregation because the fibril degradation creates new free ends that can expand to fibrils. In opposition, when they are in high concentration, the number of available functional groups to interact with the peptide is higher, which leads to a faster breakage of fibrils. If breakage is faster than the elongation, fibrils are converted to monomeric structures, hampering fibrillation. This hypothesis is validated by the ability of dendrimers to degrade pre-existent aggregates, which has been reported in several dendritic architectures [19,21,106,109,111,127,128,129]. The dendrimer/peptide ratio also influences the secondary structure and morphology of fibrils. In the case of low dendrimer/peptide ratios, the transition to β-sheet is accelerated, and fibrils are more clumped together [19,20,21]. On the other hand, at high dendrimer/peptide ratios, there is a drop or complete inhibition on the β-sheet conformational transition, and nonfibrillar/amorphous aggregates are common [19,20,107,120,126,128]. This effect indicates that the dendrimer/peptide interaction can not only change the conformation of the amyloid species, but also inhibit its aggregation by simultaneous binding to peptide monomers and blockage of the fibril ends [19].

Even though most of the described dendrimers follow the above tendency for the dendrimer/peptide ratio, that is not true for some of them. That is the case of the GATG-Morpholine and Lysine dendrimers, which follow an inverse tendency (Table 1) [110,125]. For G3 GATG-Morpholine, a low dendrimer/peptide ratio (1:5000) have no effect on the fibrillation of Aβ (1-28) while a high dendrimer/peptide ratio (1:50) led to an increase in the elongation rate and fibril number. The dendrimers at the high molar ratio accelerated the conformational transition to β-sheet, and the fibrils formed were longer, more clumped together, and higher in number [110]. The same tendency was found in the case of G5 of lysine dendrimers [125]. These results show that the interaction of the GATG-Morpholine and G5 Lysine dendrimers with the Aβ peptide did not result in fibril breakage or inhibition of fibrillation. Instead, it accelerated its aggregation demonstrating the pro-amyloidogenic properties of these dendrimers. Nonetheless, GATG-Morpholine dendrimers protected B14 cells against Aβ toxicity to a higher extent when they were in high concentration, indicating that the shift of amyloid towards fibrils could be beneficial as the most cytotoxic species are small non-fibrillar oligomers [110].

### 4.2. Functional Surface Groups

Another important aspect to consider is the charge and the nature of the surface functional groups of the dendrimer. In the first report of Supattapone et al., it was suggested that the anti-amyloidogenic effect of dendrimers was dependent on the positive charge of the -NH_2_ functional groups, as the G4 PAMAM-OH dendrimer showed no effect [102]. The described anti-amyloidogenic properties of PAMAM, PPI, CPD, VPD, and carbosilane dendrimers in Aβ, PrP, and α-synuclein reinforced that premise (Table 1). Additionally, G3-G5 carboxyl-terminated poly(amido amine) (PAMAM-COOH) dendrimers show no anti-amyloidogenic properties [107,126]. For this reason, the interaction of dendrimers and amyloid peptides was suggested to be based on the electrostatic interaction of cationic functional groups with the negative charge residues of the amyloid monomers’ amino acids. In fact, the net charge of the Aβ (1–42) peptide is −3.2 at the physiological condition (pH 7.4) [130], which can facilitate the interaction of the cationic dendrimers with the peptide. The aggregation of the peptide is also dependent on the pH, which indicates that the charge of the amino acid residues is important for the aggregation process [22]. It has been suggested by Klajnert et al. that the cationic dendrimers can interfere with the residue of Asp-7 in Aβ (1–28), inhibiting in this way the formation of a salt bridge between Asp-7 and His-13, which in turn is needed to stabilize the β-sheet aggregates [22].

On the other hand, electrostatic interaction solely cannot explain the anti-amyloidogenic properties of the dendrimers as neutral dendrimers have shown a similar ability to inhibit amyloid aggregation and degrade previously formed aggregates [20,108,111,118,119,121]. Several studies of maltose-functionalised PPI glycodendrimers (mPPI) have shown that these dendrimers can reduce the dendrimer cytotoxicity while maintaining the same anti-amyloidogenic properties of cationic PPI dendrimers. In the same manner, gallic acid-terminated GATG (GATG-Ga) dendrimers demonstrated anti-amyloidogenic properties and the ability to degrade pre-formed fibrillar aggregates [111]. As their effect was dependent on their surface multivalency, it suggests that the surface hydroxyl moieties of gallic acid play a role in the interaction with the Aβ peptide. Both GATG-Ga and mPPI glycodendrimers can interact with the peptide by nonspecific hydrogen bonds from the gallic acid and maltose units, creating peptide/dendrimer interactions with a similar extent to the electrostatic bonding by positively charged PPI dendrimers [121]. Additionally, negatively charged sulphate-terminated dendrimers also reported anti-amyloidogenic properties. Sulphate-maltose PPI (G5 mPPI S) dendrimers showed to reduce fibrillation, slow down the conformational transition of Aβ (1-40) to β-sheet, and completely avoid Aβ-related cytotoxicity [120]. Likewise, negatively charged amphiphilic polyphenylene dendrimers functionalised with sulphonic acid and n-propyl groups revealed the ability to inhibit Aβ fibrillation, disintegrate pre-formed aggregates, and prevent Aβ neurotoxicity [127]. Their neuroprotective and anti-amyloidogenic effects were linked to the direct interaction with Aβ peptide. Analogously, dPGS showed to directly interact with Aβ (1–42) peptide, specifically with Aβ oligomers [131]. This interaction impaired fibril formation, as the presence of an equimolar concentration of dPGS resulted in fewer fibrils overall, and much shorter and thinner fibrils than Aβ (1–42) incubated alone. Therefore, the anti-amyloidogenic properties of dendrimers cannot depend only on the electrostatic interactions between peptide and dendrimers, but rather on the electrostatic and nonelectrostatic interactions, such as hydrogen bonds.

Based on the bovine/human serum albumin ability to inhibit Aβ (1–42) aggregation [132,133], the group of Yan Sun proposed a hydrophobic binding-electrostatic repulsion (HyBER) hypothesis for the fibrillation inhibition of dendrimers. In this model, the Aβ peptide aggregation is inhibited by the conformational change of the peptide by both hydrophobic interactions and electrostatic repulsions, leading to off-pathway aggregation and/or the decrease of on-pathway aggregation. To prove their theory, they modified the surface of negatively charge G3-G5 PAMAM-COOH with phenethylamine (PEA) to introduce hydrophobic groups at the surface. The resulting dendrimer is a phenyl-derivatized PAMAM-COOH (PAMP) with an anionic charge and a partially hydrophobic surface. PAMP dendrimers inhibited the Aβ (1–42) aggregation and reduced Aβ cytotoxicity in SH-SY5Y cells, in a concentration-dependent way [107]. The PAMP neuroprotective and inhibitory properties were dependent on the degree of substitution of the carboxylic acid group for phenyl groups, being the optimal degree around 30%. In this case, at the equimolar dendrimer/peptide ratio, the G5 PAMP dendrimers decreased the elongation rate of Aβ aggregation and the final fibril number by 70%, and they inhibited the conformational transition to a β-sheet. The resulting aggregates were nonfibrillar but instead appear as irregular aggregates. The reported pieces of evidence were a result of the conjugation of hydrophobic binding and electrostatic repulsion since a phenyl-derivatized hydroxyl-terminated poly(amido amine) (PAMP-OH) with a 30% hydrophobic surface could not inhibit Aβ fibrillation nor attenuate its cytotoxicity [107]. The HyBER effect was shown to depend on the structure of the dendritic structure, as low PAMP generations were unable to interfere with the Aβ aggregation or had a much lower effect (Table 1) [126]. In the case of low generation, the terminal functional groups are sparse. When Aβ (1–42) interacts with them, it will bind hydrophobically to the dendrimer but may not suffer electrostatic repulsion by the negatively charged groups because they are too distant for the electrostatic repulsion to happen. Therefore, the density of surface functional groups and the proper distribution of hydrophobic groups and negative charges on the dendrimer is of great importance for the HyBER effect to happen.

The HyBER effect was also evidenced in mPPI [20,118]. To study the effect of the density of maltose units on the surface of G4 mPPI, Klementieva et al. designed a G4 of maltose-open shell PPI dendrimer (G4 mPPI OS) in which 40% of terminal amine groups were modified with maltose molecules. Consequently, this dendrimer has maltose units and -NH_2_ functional groups at the surface, allowing both nonelectrostatic and electrostatic peptide interactions to occur. G4 mPPI OS exhibited a higher fibrillation inhibitory capacity than mPPI, as shown by the lower dendrimer/peptide ratio needed to complete inhibit fibrillation (Table 1). Hence, the conjugation of electrostatic and non-electrostatic interactions between dendrimers and peptides seems to favour the inhibition of amyloid aggregation. Nonetheless, neither G4 mPPI nor G4 mPPI OS could improve memory deficits in APP/PS1 transgenic mice. Instead, the chronic administration of G4 mPPI OS led to memory impairment in wild-type (WT) mice when compared with PBS- and G4 mPPI-treated WT animals, indicating that the amino groups of G4 mPPI OS may have a detrimental effect [118].

### 4.3. Generation

The inhibitory effect of the dendrimer on fibrillation is also generation dependent. In general, the higher the generation, the lower the amount of dendrimer necessary to inhibit amyloid aggregation and disrupt already existing fibrils is (Table 1) [134]. For example, G5 PAMAM can completely inhibit the aggregation of Aβ (1–28) at the dendrimer/peptide ratio of 0.02 while the same ratio only decreases elongation and fibril amount in 65% for G4 PAMAM and 50% for G3 PAMAM (Table 1) [19]. An increase in generation translates into an increase in the size of the particle, together with an increase in the number and density of surface functional groups. On one hand, a size increase can allow dendrimers to interact with more than one peptide at once, hampering the fibrillation process [19]. On the other hand, an increase in the number of functional groups allows more interactions with the peptide, which justifies the higher inhibitory capacity of high generation dendrimers. In addition, the increase in the density of the surface functional groups can facilitate the interaction between the peptide residues and the dendrimers, resulting in a greater inhibitory capacity. Nonetheless, the increase in generation can bring no major advantages when the charge density is already high in the previous generation. In this case, the increase of functional groups will not translate into a higher number of interactions with the peptide and can instead hamper the dendrimer/peptide interaction due to the particle size increase. One example of this effect is the G5 and G6 PAMP, where both generations had similar effects on the amyloid aggregation process (Table 1) [126]. The authors also showed that the dissociation constant between Aβ (1–42) and G5 PAMP (30% Phenyl groups) was higher than the one between Aβ (1–42) and G6 PAMP (30% Phenyl groups), indicating that G5 PAMP binds more tightly to Aβ (1–42) than the G6. In this case, the generation increase has not translated into an improvement in the dendrimers’ effect, but rather led to a decrease in the dendrimer affinity to the peptide.

Low generation dendrimers can also present strong anti-amyloidogenic properties. G0 and G1 GATG-Ga dendrimers exhibited a high inhibitory effect on fibrillation of Aβ (1–42) and the disassembly ability to preformed fibrils in a concentration-dependent manner [111]. They reduced the number of small and non-fibrillar oligomeric aggregates, and elongated fibrils were replaced by shorter fibrils and unstructured aggregates (condensed and less organized). More importantly, their presence reduced the amount of Aβ aggregates in the cellular environment (both fibrillar and oligomeric forms), which translated into a neuroprotective effect on SH-SY5Y cells after Aβ (1–42) exposure. The bioactivity of GATG-Ga dendrimers was proportional to the number of gallic acid moieties, where higher multivalency (and generation) led to increased bioactivity. Nonetheless, the dendrimers with the highest multivalency (3G1-GaOH) had lower bioactivity than G0 and 2G1-GaOH, which was suggested to be linked to their higher hydrophobicity that made them more prone to interact with themselves than Aβ peptide. Analogously, cationic G0 VPD (VPD-1 and VPD-2) and G2 carbosilane dendrimers (BDBR7 and BDBR11) demonstrated a robust inhibition on α-synuclein fibrillation (over 90%) by preventing the conformational transition of the peptides to a β-sheet [112,115]. Their anti-amyloidogenic properties were stronger than G4 PAMAM and G3/G4 CPD for the same peptide [115]. In VPD and carbosilane dendrimers, each dendron possesses two cationic amine groups, which convert in a high net cationic charge despite their low generations. The high cationic net charge of VPD and carbosilane dendrimers allow them to interact with α-synuclein electrostatically, which can explain their high anti-amyloidogenic properties. In the case of GATG-Ga, each gallic acid molecule presents three hydroxyl groups, indicating that G1 GATG-Ga dendrimers (2G1-Ga-OH) possess 18 hydroxyl surface groups. Their multivalence results in a densely packed surface that allows them to create stable interactions with amyloid peptide, hampering fibrillation. Altogether, these reports reveal that low generation dendrimers can exhibit high anti-amyloidogenic properties when their surface multivalency is still high. Therefore, more important than the generation of the dendrimers is their surface multivalency, as it can strongly influence their bioactivity.

### 4.4. Topology

Lastly, dendrimers’ topology can also affect their anti-amyloidogenic properties. As different topologies can result in a different spatial arrangement of the surface functional groups, this can impact their interaction with the peptide, and consequently, impact their biological properties. Ferrer-Lorente et al. studied the effect of the dendrimers’ topology on α-synuclein fibrillation using spherical, bow-tie and dendron cationic carbosilane dendrimers, all bearing the same number of functional groups (-N^+^(Me)_3_) [113]. Bow-tie dendrimers showed a slightly higher capacity to inhibit α-synuclein fibrillation than spherical dendrimers or dendrons, even though all dendritic topologies could inhibit α-synuclein fibrillation. The same bow-tie and dendron structures also prevented the amyloid formation of amyloidogenic islet amyloid polypeptide (hIAPP) in pancreatic islets isolated from Tg-hIAPP mice and exposed to dendrimers for seven days in vitro [135]. Here, spherical carbosilane dendrimers had no effect on amyloid formation, and dendrons were the most active topology. Hence, topologies that favour clustering and a spatial multivalency of surface functional groups, such as bow-tie and dendron topologies, seem to translate into stronger anti-amyloidogenic properties.

Moreover, the effective multivalency of the dendritic structures is also an important parameter. Xiang et al. compared the anti-amyloidogenic properties of sulphonic acid/n-propyl groups (APD)-functionalised amphiphilic polyphenylene dendrons and their dendritic topology (four APD biotin-terminated dendrons assembled onto the protein streptavidin) on Aβ assemble/disassemble [127]. The polyphenylene dendrimers demonstrated a higher capacity to inhibit Aβ aggregation than their dendrons, as lower dendrimer/peptide ratios were needed to completely inhibit fibrillation (1:5 vs. 4:1) (Table 1). Moreover, dendrimers could completely disassemble preformed fibrils at the dendrimer/peptide ratio of 2:1, whereas dendrons could only accomplish it at the 4:1 dendrimer/peptide ratio. Therefore, the higher number of terminal groups/valency of dendrimers compared with dendrons translated into higher anti-amyloidogenic properties. Altogether these reports suggest that more important than the topology of dendritic structures is the effective multivalency of the surface functional groups, as it determines their effective interaction with amyloid peptides.

In conclusion, the anti-amyloidogenic properties of dendrimers depend on the dendrimer/peptide ratio, generation, and the characteristics of the dendritic surface, including its charge, nature, topology, and type/density of the functional groups. Since the Aβ peptide possesses hydrophobic, cationic, and anionic amino acids, it can interact with dendrimers in distinct ways, resulting in several possible mechanisms of fibrillation inhibition. Nonetheless, the number, density, and nature of the surface functional groups of the dendrimers are of great importance for the dendrimer’s anti-amyloidogenic properties. More densely packed surfaces with hydrophobic and/or charged groups create dendrimers more prone to inhibit amyloid aggregation and attenuate its cytotoxicity. Nonetheless, the in vitro anti-amyloidogenic properties of dendrimers can translate into no clinical relevance in in vivo models, thus analysing their effect in in vivo models is imperative.

## 5. Dendrimers as Anti-Tau Agents and Inhibitors of Acetylcholinesterase Activity

Besides the ability to inhibit Aβ aggregation, dendrimers can also inhibit the aggregation of tau, as shown by Wasiak et al. [21]. In this study, G3 and G4 CPD have been shown to interfere with tau aggregation in a generation- and concentration-dependent way. By thioflavin S (ThS) fluorescence measurements, both G3 and G4 CPD showed the ability to inhibit tau aggregation in vitro when the dendrimer/peptide molar ratio was high (1.5). At the same molar ratio, TEM images showed that the presence of CPD caused tau aggregates to become more amorphous compared to the long and fibrillar species of control tau aggregates. However, the changes in the aggregates’ morphology were dependent on the dendrimer’s generation. G3 CPD led to a substantial reduction in the formation of fibrillar structures and the resultant aggregates were mostly amorphous. Conversely, G4 CPD showed a less apparent effect on tau filamentous aggregation and its effect was only observed by a shortening of the fibrillar structures. For both CPD G3 and G4, a low dendrimer/peptide ratio (0.15) had no effect on tau fibrillation.

The same CPD have shown the ability to inhibit acetylcholinesterase (AChE) activity [24]. AChE is an enzyme involved in acetylcholine-mediated neurotransmission and signal transduction. It hydrolyses the neurotransmitter acetylcholine (ACh) to choline and acetate, preventing re-excitation after the stimulated cell [136]. In AD pathology, a loss of cholinergic neurons and cholinergic activity have been described, leading to cognitive impairment and dysfunction [137]. Hence, compounds that can inhibit or reduce AChE activity could counterbalance the ACh decrease, helping signal transduction and attenuating AD cognitive dysfunction. Because of that, most of the AD-approved drugs focus on increasing the level and action duration of ACh by inhibiting cholinesterase activity. In this study, Wasiak et al. showed that CPD also has this ability. Both G3 and G4 CPD affected the AChE activity on N2a cells in a concentration-dependent way. CPD inhibitory activity cannot be linked to their antagonistic ability since they have no structural similarity to ACh and are much larger than ACh (4.2 nm to 5 nm for G3 and G4 vs. 1.8 m for ACh). Instead, it has been suggested that the dendrimers change the conformation of the protein not by direct interaction with the catalytic pocket of the enzyme but rather by modifying the membrane fluidity or interaction with other protein components of the membrane.

AChE inhibitory activity has also been described in PAMAM dendrimers [23,25]. In a study by Klajnert et al. [25], G4 PAMAM and G4 PAMAM-OH presented a biphasic effect on the activity of membrane-bound AChE. At low concentrations, dendrimers caused a significant increase in AChE activity, whereas at high concentrations they inhibit its activity. Even though both dendrimers could inhibit AChE activity, their effect was distinct, as the maximum activation occurred at 25 µM for G4 PAMAM-OH and 100 µM for G4 PAMAM. The authors suggested that the inhibitory effect of both PAMAM dendrimers on AChE activity was consistent with an uncompetitive inhibition and could be explained by direct interaction between dendrimers and the enzyme and/or indirect effect via membrane condition modifications, which have previously been described to affect AChE activity (e.g., membrane fluidity) [138]. To elucidate PAMAM’s mechanism of AChE inhibition, the same authors studied the effect of PAMAM dendrimers on pure AChE [23]. Here, G4 PAMAM, G4 PAMAM-OH, and G3.5 PAMAM-COOH showed to reduce AChE activity and directly interact with the protein. Their data indicated that PAMAM dendrimers change AChE activity by changing its conformation and catalytic activity. They suggested that dendrimer-AChE interaction is based on electrostatic and hydrophobic interactions, resulting in either blockage of AChE gorge or AChE conformational change.

Lastly, AChE inhibitory properties have been described in VPD [139]. In this study, they tested the effect of two types of VPD dendrimers on the activity of AChE and butyrylcholinesterase (BChE), either in its pure form or membrane-bound. They found that the smaller and less toxic VPD dendrimer (dendrimer 1, benzaldehyde-terminated VPD) could reduce AChE and BChE activity in a concentration-dependent way for both free and membrane-bound proteins. However, they did not induce a conformational change in the cholinesterases. Hence, the authors suggested that these dendritic structures could reduce AChE and BChE activity by binding to the peripheral sites of both enzymes and inhibiting their catalytic activities. Conversely, a slightly bigger and more toxic VPD dendrimer (dendrimer 2, diethyl phosphite-terminated VPD) reduced AChE and BChE activity in a concentration-dependent way, accompanied by protein conformational changes. In this case, dendrimers 2 seem to inhibit cholinesterase activity by changing their conformational.

## 6. Dendritic Structures as Anti-Inflammatory Agents

Over the last decades, the importance of inflammation in AD pathology has become clear. First, several reports demonstrated that chronic neuroinflammation is present in AD pathology. For example, microglia and astrocytes, which are the central nervous system (CNS) resident immune cells, are expanded in AD and are associated with a pro-inflammatory profile (M1 phenotype) [140]. Second, mutations on the triggering receptor expressed on myeloid cells 2 (TREM2) and myeloid cell surface antigen CD33 increase the susceptibility for AD pathology [141,142,143]. As both genes are highly expressed in monocytes, macrophages, and microglia, they represent a link between AD and inflammatory cells. Finally, epidemiological studies showed that the long-term use of nonsteroidal anti-inflammatory drugs (NSAIDs) for the treatment of chronic inflammatory diseases, such as rheumatoid arthritis, resulted in a 50% reduction in the risk of developing AD [144,145]. Hence, inflammation has a major impact on AD pathology.

Microglia, the principal immune cells in the CNS, have a dichotomous role in AD pathology. Microglia are myeloid cells that are responsible for the CNS surveillance and clearance of pathogens, damaged tissue, and synapses. As intracellular Aβ deposits have been observed in microglia in AD brains, they can phagocyte and degrade Aβ aggregates, hence contributing to the clearance of Aβ [146]. Additionally, post-mortem studies showed that activated microglia are in proximity to amyloid plaques and NFTs, demonstrating that they react with the protein aggregates [147]. With the progression of the disease, they become chronically activated and represent a harmful player in AD [140]. Their activation leads to the release of pro-inflammatory cytokines, such as IL-1α, IL-1β, IL-6, TNF-α, and other [140]. These cytokines, in turn, downregulate Aβ phagocytic receptors and Aβ degrading enzymes [148],increase the production of APP, and enhance the activity of the APP cleavage enzymes (γ- and β-secretase) [149,150,151,152], translating in higher Aβ accumulation. IL-6 have also been suggested to increase tau phosphorylation, exacerbating AD pathology [153]. Additionally, M1-activated microglia release ROS, neurotoxins, and others, leading to a neurotoxic effect.

Interestingly, increasing evidence suggests that systemic inflammation has an impact on neuroinflammation and AD pathology [154]. For example, chronic pro-inflammatory diseases like diabetes and obesity have been associated with a higher cognitive decline over the years and are risk factors for dementia/AD [155,156,157]. Peripheral inflammation results in innate immune system activation, leading to pro-inflammatory cytokine production. The blood circulation of these cytokines in turn can affect neurons and glial cells and promote the extravasation of peripheral immune cells to the brain, translating into neuroinflammation [154]. Bacterial respiratory infection has been shown to promote T cell infiltration to the brain of APP/PS1 mice, which led to increased glial activation and Aβ deposition [158]. Therefore, therapeutic strategies that can modulate the inflammatory response, either systemic or neural, could have a beneficial impact on AD pathology.

In the context of inflammation, dendrimers were first explored as nanocarriers of NSAIDs. However, it was not soon after, that different dendritic structures showed anti-inflammatory properties per se (Table 2). The first report dates to 2004, where a G3.5 carboxyl-terminated PAMAM with surface glucosamine residues inhibited the production of pro-inflammatory cytokines and chemokines from lipopolysaccharide (LPS)-induced macrophages and dendritic cells [159]. Yet, unmodified PAMAM dendrimers have also shown similar anti-inflammatory properties [160]. Through three independent inflammation models in rat, Chauhan et al. demonstrated that intraperitoneally injected PAMAM dendrimers inhibited the inflammatory response in a dose- and time-dependent manner, which in some cases was stronger than the NSAID indomethacin alone. Their anti-inflammatory effect was linked with the inhibition of cyclooxygenase (COX)-1/-2 and nitric oxide (NO) release in rat peritoneal macrophages. In the same manner, other dendritic structures, namely phosphorous-containing dendrimers [161,162] and dPGS [163,164], demonstrated anti-inflammatory properties. Polyphosphorhydrazone (PPH) dendrimers, namely G1 azabisphosphonate (ABP)-terminated PPH dendrimers, exhibit immune-modulator and anti-inflammatory properties by the alternative activation of human monocytes into an anti-inflammatory phenotype in vitro, which in turn increases IL-10 secretion by CD4^+^ T cells [165]. These dendritic structures also inhibited the proliferation of IL-2-stimulated/pro-inflammation CD4^+^ T cells and the maturation of human monocyte-derived dendritic cells in vitro, thereby controlling the inflammatory response [166,167]. The oral or intravenous administration of ABP-terminated PPH dendrimers in a mouse model of rheumatoid arthritis (IL-1-ra^−/−^ mice) led to a drastic decrease in the serum levels of pro-inflammatory cytokines (IL-1β, IL-6, and IL-17) and metalloproteins (MMP-3 and MMP-9), which translated in the control of the disease progression and clinical symptoms [168,169]. From the same family of dendrimers, the G3 and G4 methoxy derivatives of PPH dendrimers (48 and 96 terminal bisphosphonate groups, respectively) showed to polarize macrophages from the pro-inflammatory M1 subtype to the anti-inflammatory M2 subtype, both in vitro and in vivo [170]. Hence, PPH dendrimers can modulate the immune response by the modulation of inflammatory cells. On the other hand, dPGS revealed anti-inflammatory properties through the prevention of massive efflux of leukocytes to the inflammation tissue. They can bind to L-selectin, P-selectin, and complement factors C3 and C5, preventing in this way leukocyte extravasation [163,171,172]. More recently the group developed a biodegradable dPGS that present similar binding properties to selectins and complement factors as non-biodegradable dendrimers [173,174].

Besides their anti-inflammatory properties, dendritic structures seem to have an intrinsic targetability to the neuroinflammatory area and microglial cells in particular (Table 2). Several reports in different neuroinflammation-associated disease models demonstrated that the local and systemic administration of G4 PAMAM-OH dendrimers resulted in a differential and increased brain uptake in neuroinflammation-associated animals, compared to age-matched healthy animals [82,83,175,177,186]. Since no appreciable differences in the dendrimer’s clearance and accumulation in other major organs were seen between control and neuroinflammation-associated animals, the increased brain uptake was directly correlated with neuroinflammation [175,177,187]. Dendrimer neural uptake is (1) inflammation-site specific, (2) proportional to the severity of the disease, and (3) dependent on the severity of the inflammation, as the dendrimer accumulation was only present in regions with BBB impairment and glial cell activation [175,188,189]. More importantly, G4 PAMAM-OH dendrimers were shown to selectively accumulate in activated microglia, astrocytes, and injured neurons, both in small [82,83,175,177,186,190] and large in vivo neuroinflammation-disease models [187,188]. For all cell types, dendrimer accumulation correlates with the injury site and injury severity, yet it varies according to the cell types [190]. Microglia is the major responsible for the dendrimer uptake, with 60–80% uptake 24h after in vivo injection in a hypoxic–ischemic neonatal mice model [190]. On the other hand, astrocytes and neurons have a much lower dendrimer uptake, with an 8–15% and 2–4% uptake, respectively [190]. The enhanced accumulation in microglial cells has also been described in dPGS (Table 2). Incubation of fluorescently-labelled dPGS in mouse organotypic hippocampal slice cultures revealed low or no dendrimer uptake in neurons and astrocytes, yet microglial cells had a strong fluorescence signal, regardless of their activation state [164]. Nonetheless, PAMAM-OH were especially taken up by activated microglia in a much faster and more extensive way than resting microglia [190,191,192,193].

The enhanced uptake of dendritic structures by microglial cells in the neuroinflammation area can be explored to deliver therapeutics in a specific and targeted way, decreasing in this way the off-set side effects. One example is the delivery of tesaglitazar using PAMAM-OH [194]. Tesaglitazar (Tesa) is a potent PPARα/γ dual agonist which exhibits an anti-inflammatory effect and can induce the polarization of microglia and macrophages to an M2/anti-inflammatory profile [195,196,197]. In this report, DeRidder et al. covalently attached ten Tesa molecules to the surface of the G4 PAMAM-OH and tested the ability of the nanoconstruct (D-Tesa) to modulate the inflammatory response in LPS-activated BV2 microglial cells. A 48h D-Tesa treatment decreased the secreted NO and inducible nitric oxide synthase (iNOS) mRNA levels, and significantly increased the mRNA levels of anti-inflammatory cytokines IL-4, IL-10, and TGF-β1, compared to LPS-only treated BV2 cells. More importantly, D-Tesa shifted the phenotype of microglial cells from an M1 to an M2 profile, as the mRNA levels of M2 markers (Arginase 1, CD206, Ccl1, and TLR28) significantly increased compared to LPS-treated cells. Tesa alone did not significantly increase the expression of these markers. Furthermore, D-Tesa significantly boosted the expression of Insulin degrading enzyme (Ide), MMP9, and CD86, translating into an increased clearance and phagocytosis of Aβ peptide. Therefore, the microglial targeted delivery of Tesa by PAMAM-OH dendrimers could not only modulate their profile to an M2 anti-inflammatory phenotype, but also improve their ability to eliminate extracellular Aβ. G4 PAMAM-OH has also been explored to deliver several other anti-inflammatory drugs to the CNS, such as NAC, which was tested in clinical trials for severe COVID-19-associated inflammation (NCT04458298) [87,198].

The combination of the enhanced uptake of the dendritic structures and their affinity to microglia with their anti-inflammatory properties can result in a targeted and anti-inflammatory therapeutic approach. Maysinger et al. showed that a 3h pre-treatment of dPGS decreased the release of NO and pro-inflammatory cytokines (TNF-α, IL-6) in 24 h LPS-exposed mice organotypic hippocampal slices compared to non-treated LPS-exposed slices [164]. dPGS could also significantly reduce LPS-induced microglial activation (but not astrocyte activation) in a concentration and time-dependent manner in vivo [180]. Hence, dPGS seems to decrease the activation state of microglia and modulate the microglial phenotype to a more anti-inflammatory profile. Additionally, dPGS acts as a scavenger for IL-6 and LCN2, which modulates the microglia crosstalk with other neuroglial cells and the activation of astrocytes [180,199]. More importantly, dPGS pre-treatment could also prevent synaptic loss in LPS-exposed slices by avoiding spine loss in CA1 neurons [164]. Therefore, the microglial uptake of dPGS can not only reduce the inflammation markers and modulate the microglia phenotype, but also have an indirect neuroprotective effect on neurons and synapses. The same synapse protective behaviour was evident in Aβ (1–42)-exposed slices [131]. Two days of exposure to Aβ (1–42) led to a significant decrease in the total dendritic spine number in the organotypic hippocampal cultures, with a notable decrease in the thin and “mushroom” spine population. The dPGS presence avoided these morphological changes on postsynaptic dendritic spines and decreased the amount of Aβ internalized by neuroglia. Their protective effect was linked to their direct interaction with Aβ (1–42) species, in a weak- and specie-specific way, and with the modulation of microglial activation. Similarly, G3 and G4 CPD decreased the TNF-α release in LPS-activated BV2 cells to the same extent as NAC, demonstrating their immunomodulating capacity [24].

Understanding which are the characteristics that grant dendrimers their anti-inflammatory properties and enhanced uptake by microglia/neuroinflammation allow us to finetune the design of new neural-targeted and anti-inflammatory dendrimers. Here, key aspects to keep in mind are the surface functionality, the generation/size and the internal chemical structure of the dendritic structure. The influence of these characteristics will be discussed in the following paragraphs.

### 6.1. Functional Surface Groups

The ability of dendrimers to reach the neuroinflammation site and target inflammatory cells depends on their surface functionalities. Nance et al. explored a newborn rabbit model of maternal inflammation-induced Cerebral Palsy (CP) to investigate the impact of surface functionality on dendrimer brain uptake by activated microglia [175]. In this in vivo model, an intrauterine injection of LPS near-term in pregnant rabbits leads to a CP phenotype and robust microglial activation in the periventricular regions of the newborn brain. On postnatal day 1, the CP rabbit kits were injected intravenously with G4 PAMAM-OH, G4 PAMAM-NH_2_, or G3.5 PAMAM-COOH (55 mg/Kg) and euthanized at 0.5 h, 4 h, or 24 h after dendrimer administration. Analysis of the affected areas revealed a differential brain uptake depending on the dendrimer’s surface characteristics. While PAMAM-NH_2_ were not found outside blood vessels, PAMAM-OH and PAMAM-COOH were found inside microglial cells, yet in distinctive time points. PAMAM-OH extravasated and localized in activated microglia at 4 h, whereas PAMAM-COOH only co-localized with microglia at the 24 h time point. The delayed uptake of G3.5-COOH suggests that a neutral surface functionality may be advantageous for rapid escape from blood vessels. Additionally, neutral charge also facilitates mobility within the brain parenchyma, as only neutral PAMAM-OH dendrimers could be found several millimetres away after an intraparenchymal injection. A novel G2 hydroxyl-terminated polyethylene glycol (PEG)-based dendrimer bearing sixty densely packed hydroxyl groups at its surface (PEGOL-60) also showed the ability to co-localize with activated microglia in neuroinflammation sites after an intravenous administration in CP kits [178]. PEGOL-60 uptake was time-dependent and significantly increased in CP kits compared to health-matched controls (~10-fold). Taking these results together, they suggest that neutral hydroxyl groups help dendrimers to target the neuroinflammation site.

The surface functionalization of dendrimers also influences their uptake at the cellular level. In serum-free conditions, treatment of BV2 murine microglia with fluorescently labelled G4 PAMAM-OH, G4 PAMAM-NH_2_, or G3.5 PAMAM-COOH (with or without LPS) led to a differential in vitro cellular uptake depending on the surface groups [176]. Neutral and anionic dendrimers exhibited similar microglial uptake while cationic dendrimers showed a 2-fold lower uptake than neutral PAMAM-OH dendrimers. LPS stimulation increased the cellular uptake for all dendrimers. Macrophage-differentiated THP-1 cells exposed to neutral hydroxyl-terminated dendritic polyglycerol (dPG) or negatively charged dPGS showed a higher particle uptake for negatively charged dPGS than neutral dPG for similarly sized particles [181]. The preferential uptake of negatively charged dendrimers by differentiated THF-1 cells was suggested to be linked to the uptake by scavenger receptors class A (SR-A), responsible for the detection and phagocytosis of charged NPs. Therefore, neutral or negatively charged dendrimers are more likely to target immunological cells.

The anti-inflammatory properties of dendritic structures are also strongly influenced by their surface functionality. In the case of G4 PAMAM, amine- and hydroxyl-terminated dendrimers showed a higher anti-inflammatory effect in carragen-induced paw edema than the carboxylate-terminated counterparts [160]. The same was true for the inhibition of nitrite formation and COX-1/COX-2 activity, where PAMAM-COOH displayed a lower capacity to inhibit nitrite formation and low to no activity towards COX-1/-2. Other reports reinforce the intrinsic anti-inflammatory properties of hydroxyl-terminated dendrimers. LPS-induced BV2 microglial cells treated with PEGOL-60 showed a reduced expression of pro-inflammatory factors (TNF-α, IL-6, IL-10, and iNOS) and an increased expression of anti-inflammatory ones (CD206, Arg1, and IL-4), comparing to LPS-only treated BV2 cells. PEGOL-60 treatment also resulted in a significant reduction in the levels of extracellular TNF-α and nitrite ions, and increased microglial viability to an oxidative insult (500 µM H_2_O_2_) [178]. Likewise, small-sized hydroxyl-terminated “click” dendrimers exhibited anti-inflammatory and antioxidant properties in LPS-induced N9 microglia [179]. Their properties were suggested to be related to the direct interaction with iNOS and COX-2. However, other reports showed that hydroxyl-terminated dendritic structures could be less effective in an inflammatory insult. While sulphate-terminated dPGS treatment decreased inflammatory markers and exhibited a neuroprotective action upon an LPS or Aβ (1–42) insult, the treatment with neutral hydroxyl-terminated dPG have no effect [131,164]. Moreover, the intranasal administration of dPG could not suppress the LPS-induced effects on microglia in vivo while dPGS administration decreased the activation state of microglia and modulated the microglial phenotype to a more anti-inflammatory profile [180]. Therefore, sulphate groups seem to have greater anti-inflammatory activity than hydroxyl ones.

Another surface functionality that has shown immunomodulating properties is the azabisphosphonate (-N(CH_2_P(O)(OH)(ONa))_2_) (ABP). As previously mentioned, ABP-terminated PPH dendrimers can alternatively activate monocytes into an M2-like phenotype and modulate the macrophage M1/M2 phenotype balance. Their anti-inflammatory assets are dependent on the presence of surface phosphonic groups, as PPH dendrimers bearing the ABP group activated human monocytes to a much higher extent than dendrimers capped with carboxylic acid groups [182]. The neutral high-generation phosphorus dendrimer (-N(CH_2_P(O)(OCH_3_)_2_)_2_) also demonstrated anti-inflammatory properties in vivo. After intravenous administration in a model of sub-chronic inflammation, these dendrimers reduced the nitrite levels, the number of migrating inflammatory cells and the expression of iNOS in migrating cells while increasing the expression of anti-inflammatory marker CD163 [170]. As both neutral and negatively charged ABP dendrimers demonstrated anti-inflammatory properties, ABP’s anti-inflammatory effect is not related to its negative charge, but rather to its surface moieties. Interestingly, ABP-terminated PPH dendrimers, but not azamonophosphonate-terminated, showed anti-inflammatory properties in IL-1-ra^−/−^ mice [169]. Therefore, not only the type of surface functional groups, but also the number/valency of phosphonate groups is an important consideration in the activity of PPH dendrimers.

### 6.2. Generation/Size and Multivalency

Dendrimers’ cellular and brain uptake is also influenced by their generation/size. Comparing G4 PAMAM-OH and G6 PAMAM-OH, G6 PAMAM-OH brain uptake was higher than G4 PAMAM-OH in neuroinflammation-associated in vivo models [175,188,193]. G6 PAMAM-OH dendrimers showed increased circulation time compared to G4 dendrimers [175,188,193], which was associated with lower renal clearance and higher reticuloendothelial system clearance in a large animal (dog) of hypothermic circulatory arrest [188]. However, G6 PAMAM-OH dendrimers showed a 4-fold lower cellular uptake than G4 PAMAM-OH dendrimers in BV2 cells [176]. Hence, the increased brain uptake of higher generations of PAMAM dendrimers is likely a result of the increased blood circulation rather than the increase in microglial uptake due to its higher size or valence. Nevertheless, macrophage-like differentiated THP-1 exposed to dPG or dPGS showed an increase in cellular uptake with particles’ growing size, regardless of the dendrimer charge [181]. As the increase in generation and size is associated with a higher number of terminal groups and multivalency, the increment in cellular uptake could be linked to the higher valency for cellular interaction. Nonetheless, understanding how the dendrimer interacts with cells and how they are transported into the cell is of extreme importance for defining the generation/size and type/number of functional groups.

An increase in generation/size has also been associated with stronger anti-inflammatory properties of dendritic structures. G4 neutral ABP dendrimers showed slightly higher anti-inflammatory activity than G3 dendrimers in vivo [170]. Likewise, G6 of amine-terminated PAMAM dendrimers presented higher COX-2 inhibition compared to amine-terminated G4 and G5, indicating that higher generation can increase the anti-inflammatory capacity of these dendrimers.

More important than the generation/size is the multivalency of the dendritic structures. In the case of the G1 PPH dendrimers, the authors synthesized and analysed the activity of several G1 PPH dendrimers bearing a different number of ABP moieties (2, 4, 8, 10, 12, 16 or 30), in vitro and in vivo, to understand the required number of active surface groups for an anti-inflammatory profile [183,184]. In vitro, the most active dendrimer to alternatively activate monocytes was 8-ABP, bearing 8 ABP groups at its surface. The presence of a higher number of ABP groups resulted in a slightly lower (yet comparable) activation of monocytes. Nonetheless, when the surface density was higher than 12 groups no differences were seen in the anti-inflammatory effects. A lower valence of ABP groups (2 or 4 ABP groups) drastically decreased the anti-inflammatory activity of the dendrimers [183]. When these dendrimers were injected intravenously in an arthritis mice model (K/BxN serum transfer), only PPH dendrimers bearing 10 or 12 ABP moieties could prevent inflammation and reduce the arthritis clinical manifestations [184]. The differential outcomes between in vitro and in vivo results were linked to the three-dimensionality of PPH dendrimers. With exception of the 16-ABP, the ABP dendrimers were intrinsically directional, with the active groups gathered on one side. However, dendrimers containing a low number of ABP groups (2-ABP and 4-ABP) had low surface crowding with free movement of the surface groups, limiting their possibility to establish stable interactions. On the other hand, dendrimers 8-ABP, 10-ABP and 12-ABP were more rigid and the ABP groups were packed on one side of the dendrimers, increasing their multivalency and allowing stable interactions to be formed. Differently, the 16-ABP dendrimer was non-directional, which means that the number of ABP groups clustered at its surface was much lower than in 10-ABP or 12-ABP, explaining their ineffective activity in vivo. 8-ABP inactivity in vivo was related to the dendrimer degradation through the hydrazone group, which results in an inactive counterpart [184]. Therefore, the bioactivity of dendrimers is more likely determined by the effective spatial multivalency of surface groups than the real number of such groups.

### 6.3. Internal Structure

Finally, the anti-inflammatory activity of dendrimers can also be influenced by the internal structure of the dendrimers. Caminade et al. synthesized and tested the biological activity of seven families of dendrimers (PAMAM, PPI, carbosilane dendrimers, PLL, and three types of phosphorus-containing dendrimers) bearing ABP groups at their surface [185]. Even though all dendrimers had the same surface functionality, only phosphorous-containing and poly(carbosilane) dendrimers could alternatively activate monocytes into an anti-inflammatory profile. The in vitro activity of dendrimers was not linked to their number of groups nor generation/size, but rather to their molecular directionality and hydrophilicity/hydrophobicity (Figure 5). Active dendrimers were overall more hydrophobic and had a directional architecture in equilibrium, with all surface functions gathered into ‘clusters’ while non-active dendrimers assume a more symmetric configuration. As the internal structure of the dendrimers influences the space architecture and distribution of terminal active groups, it plays a critical role in the biological activity of dendrimers.

In conclusion, dendrimers and dendritic polymers can target neuroinflammation and inflammatory cells, while possessing anti-inflammatory properties on their own. Such characteristics can be explored to deliver therapeutics in a specific and targeted way to inflammatory cells or as a multifunctional therapeutical approach. These characteristics are largely influenced by their chemical and structural features, namely their surface functionality, size/generation, and internal structure. Based on the current literature, higher generation and neutral or negatively charged dendritic structures seem more likely to target microglial cells and neuroinflammation site. Their anti-inflammatory character is strongly influenced by the surface functionalities, being neutral (-OH) or negatively charged (-N(CH_2_P(O)(OH)(ONa))_2_ and -SO_3_^−^) dendritic structures more likely to exhibit anti-inflammatory properties. The number/valency of such groups is of extreme importance since it determines the establishment or not of stable cellular interactions. By increasing the generation of the dendrimer, one increases its surface multivalency, which in turn enhances its anti-inflammatory properties. Nevertheless, it is crucial to understand how the dendrimer/dendritic nanoparticle interacts with cells, as the increase of surface moieties may not convert into a higher biological activity, but rather may be detrimental. Finally, the internal structure of the dendrimers was shown to strongly influence their properties as it can change the spacial architecture and distribution of terminal active functional groups. Here, architectures that allow a packed multivalence surface are necessary to ensure activity.

Of notice, some dendrimers that have been described to possess anti-inflammatory properties have also been associated with a pro-inflammatory effect. G3 PAMAM dendrimers were shown to enhance the infiltration of leukocytes in the mouse air pouch model [200]. In the same manner, the intravenous administration of anti-inflammatory G1 ABP-terminated PPH dendrimers in healthy non-human primates increased the level of C-reactive protein and aspartate- and alanine-amino transferases, all liver enzymes produced in the context of inflammation [201]. The same anti-inflammatory/pro-inflammatory duality is reported in inflammation itself. For example, IL-10 is an antioxidant cytokine produced by CD4^+^ Th2 cells, monocytes, and B-cells, which inhibits the expression of Th1 pro-inflammatory cytokines, such as IL-2 and IFN-γ. Nonetheless, IL-10 overproduction has been related to the severity and fatal outcome in sepsis, demonstrating the “double-edged sword” behaviour of cytokines and inflammation [202]. As inflammation itself, the anti-inflammatory effect of dendritic structures seems to vary according to the context and pharmaceutical formulation in which they are administrated and present a dual behaviour towards inflammation. Hence, the properties of dendritic structures in inflammation should be carefully analysed for the application they were designed.

It is worth noting that dendrimers’ pro-inflammatory activity can be avoided by prevention of immune system recognition. One of the most explored approaches for immune system circumvention is the functionalisation of the surface of dendrimers with PEG. PEGylation can prevent the opsonization of nanoparticles by complement peptides and immunoglobulins, avoiding immune recognition by dendritic cells, blood monocytes, granulocytes, and macrophages [203]. PEGylation can also reduce the systemic toxicity of nanoparticles and improve their circulation time and mask the nanostructures from the immune system [203]. Besides PEG, other polymers like poly(glycolic acid) (PGA), poly(glycerol) (PG), and poly(N-(2-hydroxypropyl) methacrylamide) (HPMA) are also being explored as functionalization moieties to avoid immune activation and recognition [204]. Therefore, even though dendrimers can trigger a pro-inflammatory response, there are strategies that can overcome the immune system activation to avoid this response.

Based on the evidence gathered so far, the anti-inflammatory properties of dendritic structures can be considered a valuable characteristic of these macromolecules that can act as a drug per se.

## 7. Dendrimers as Antioxidants and Chelators

There is strong evidence that oxidative stress is elevated in AD. Several studies have demonstrated that lipid peroxidation [205,206,207,208,209], protein oxidation [210,211], and DNA damage and fragmentation [212,213] were increased in AD and MCI brains, particularly in regions where NFT and Aβ plaques were present. As these phenomena are related to the presence of oxidative species, it indicates that there is excessive production of ROS in AD. Aβ oligomers were shown to generate ROS by the direct activation of the NADPH oxidase [214]. Additionally, they can interact with NMDA receptors and destroy their activity, resulting in the generation of extracellular ROS and excessive Ca^2+^ ions influx into neurons, which in turn cause mitochondrial dysfunction [214]. Aβ oligomers’ direct interaction with mitochondria can also lead to mitochondrial dysfunction [45]. Since the mitochondria are not only a source of ROS, but are also a reservoir of Ca^2+^ ions and apoptotic proteins, their dysfunction has severe implications for neuronal survival and neurotoxicity [215]. Other disease hallmarks, such as tau hyperphosphorylation and inflammation, are also a source of ROS, hence the high generation of ROS in AD is likely to surpass the antioxidant capacity of cells.Strategies that can reduce the elevated ROS are potent neuroprotective strategies.

Some types of dendrimers have also been shown to possess antioxidant properties. As aforementioned, PAMAM, PEGOL-60, PPH, and other hydroxyl- and sulphate-terminated dendritic structures can reduce the presence of NO and decrease iNOS expression [160,164,170,178,179]. In the same manner, VPD decreased ROS levels and increase the catalase activity in mouse mHippoE-18 cells, compared to non-treated cells [116]. Hence, these dendrimers could decrease cellular oxidative stress by modulation of the expression of oxidant or antioxidant enzymes.

Several dendrimers have demonstrated antioxidant properties by the scavenger capacity of oxidant species as well. The scavenger properties of dendrimers have been described in PAMAM [216], glycodendrimers [217,218], CPD [24], and original dendrimers [219,220,221,222,223,224,225]. These dendrimers were based on the presence of structural moieties known to possess antioxidant properties, such as carbazole [219,226], triazole [217,227], Meldrum’s acid derivatives [220], gallic acid [228,229,230,231,232,233], and other polyphenol compounds [234,235,236,237]. All dendrimers showed the ability to scavenge oxidative species like 1,1-diphenyl-2-picrylhydrazyl (DPPH), hydrogen peroxide (H_2_O_2_), and hydroxyl radical. One example of these properties is the report of del Olmo et al. [228]. In this study, G1 and G2 carbosilane dendrimers were functionalised with ferulic acid, caffeic acid, or gallic acid. Ferulic acid (3-(4-hydroxy-3-methoxyphenyl)-2-propenoic acid), caffeic acid (3-3,4-dihydroxyphenyl)-2-propenoic acid), and gallic acid (3,4,5-trihydroxybenzoic acid) are natural polyphenols reported to act as radical scavengers, have anti-inflammatory activity, and activate antioxidant enzymes [238]. All carbosilane dendrimers successfully scavenge DPPH to a stronger extent than the free polyphenols. Gallic acid-functionalised carbosilane dendrimers exhibited the strongest DPPH activity and ferulic acid-functionalised counterparts exhibited the weakest, indicating that the number and position of the hydroxyl groups impact the antioxidant activity of the dendrimers. The ability of polyphenolic dendrimers to reduce Fe(III) to Fe(II) was also evaluated by a ferric-reducing antioxidant power (FRAP) assay. Caffeic acid- and gallic acid-dendrimers showed the capacity to reduce Fe(III) ions to a similar or higher extent (respectively) than their respective free polyphenols. Nonetheless, G1 ferulic acid-functionalised carbosilane dendrimers demonstrated a lower capacity than ferulic acid alone. Gallic acid-functionalised dendrimers demonstrated the highest antioxidant activity of them all. The antioxidant capacity of these dendrimers was not influenced by generation, as G2 did not exhibit a superior antioxidant activity than G1. More importantly, these dendrimers demonstrated biocompatibility in Neonatal Human Foreskin fibroblasts (HFF-1), indicating that their use in biomedical applications is possible.

The presence of reduced thiol groups has also been explored in dendrimer synthesis as an antioxidant moiety. G4 PEGylated cysteine-modified lysine dendrimers with multiple reduced thiols exhibited scavenging activity towards H_2_O_2_ and hydroxyl radicals [239]. Their scavenging activity normalized the glutathione levels and prevented the increase of plasma alanine aminotransferase (ALT) activity in hepatic ischemia/reperfusion in mice, indicating that they could inhibit the hepatic injury caused by ischemia/reperfusion. L-cysteine and glutathione in the same thiol concentration could not prevent hepatic injury. In the same manner, L-cysteine and L-serine modified G3 PAMAM dendrimers with multiple reduced thiols were also shown to scavenge DPPH, H_2_O_2_, and hydroxyl radical and to prevent renal ischemia/reperfusion injury in mice [240].

Dendrimers have also been used to mimic cellular antioxidants, such as catalase. Catalase is a metalloprotein that is responsible for the conversion of H_2_O_2_ to water and oxygen, eliminating in this way oxidative stress [73]. Wang et al. have reported the capacity of Acetylated-G9 PAMAM dendrimers encapsulating platinum (Ac-G9/Pt) to mimic this enzyme in size, shape, and function [73]. Ac-G9/Pt nanoparticles demonstrated a relatively lower affinity to H_2_O_2_ than catalase, yet they degraded H_2_O_2_ at the same rate as catalase. When Ac-G9/Pt was added together with H_2_O_2_ to cells, it could attenuate the oxidative stress of H_2_O_2_ to the same extent as catalase, demonstrating their catalase-like activity in vitro. They also exhibit the capacity to scavenge DPPH.

Lastly, dendrimers have also been described as chelators. Metal ions like iron, zinc, and copper have been linked with AD pathology. They are abnormally elevated in AD patients and are more concentrated in Aβ plaques due to coordination with Aβ [241]. They are thought to accelerate Aβ aggregation, exacerbate oxidative stress, disturb normal neurological, and impair cognitive cognition [58]. Hence, metal-targeted chelation compounds could represent an AD therapeutic opportunity. Several types of dendritic structures have demonstrated the ability to act as chelators. This effect has been described on PPI [242,243], maltose-shell PPI [120,243,244], carbosilane dendrimers [245], and others. A recent example of the chelator effect on AD pathology is the work of Janaszewska et al. [120]. In this study, the effect of a G4 mPPI containing sulphate groups (G4 mPPI S—0.76 equivalents of hydroxyl groups of maltose units were a -OSO_3_^−^ unit) on Cu(II)-induced Aβ fibrillation was studied. Results showed that these dendrimers could largely avoid the Cu(II)-induced Aβ aggregation and inhibit its conformational transition to β-sheet. This effect was associated with the binding of G4 mPPI S to the amyloid peptide (as previously described) and the complexation of the ions at the dendrimer/peptide interface. By electron paramagnetic resonance (EPR) spectroscopy, G4 mPPI S was shown to complex with Cu(II), yet differently depending on the Cu(II)/dendrimer ratio. At the lowest Cu(II)/dendrimer molar ratios, Cu(II) was complexed by the internal-dendrimer nitrogen sites. After saturation of these sites, Cu(II) binding with sulphate groups occurred.

## 8. Dendrimers as Nanocarriers

In addition to their beneficial properties to act as a drug per se, dendrimers are also exceptional carriers due to their intrinsic characteristics. Their well-defined and highly tuneable structure makes them capable of carrying any kind of therapeutics. Dendrimers can carry therapeutics in three distinct ways: (1) covalently bound to their surface functional groups; (2) trapped within their internal cavities, called a “dendritic box”; or (3) complexed by non-covalent interactions formatting a dendritic nanoparticle. The preferred method of cargo can be chosen depending on the nature and/or size of the therapeutic.

Dendrimers have been widely used as a delivery system in neurodegenerative diseases [14,17]. In the AD field, they have been explored to carry either AD therapeutics [246,247,248,249,250] or other compounds that can alleviate the pathology. Here, preclinical studies have demonstrated their use to hamper amyloid burden [251,252], tau pathology [253], oxidative stress, and inflammation [194,254,255,256,257,258].

Most of the drugs approved for AD treatment are orally available. That is the case for AChE inhibitors (donepezil, rivastigmine, and galantamine) and NMDA receptor antagonists (memantine). Nevertheless, they present a low brain permeability. To reach the necessary cerebral concentrations, high dosages are usually needed, which can result in undesirable side effects. Therefore, these drugs are good candidates to be transported by a drug delivery system to improve their pharmacokinetic and pharmacodynamic properties while reducing their offsite side effects. Lactoferrin (Lf)-conjugated G3 PAMAM dendrimers have been explored as carriers of rivastigmine [246,247] and memantine [248]. Both rivastigmine-encapsulated PAMAM-Lf and memantine-loaded PAMAM-Lf reduced the toxicity of the free drug, induced an improvement in its pharmacokinetic properties and increased its brain uptake. More importantly, intravenous administration of these nanoparticles improved object recognition and spatial memory in AD-induced mice (Scopolamine-induced and AlCl_3_-induced AD animal models) to a stronger extent than the free drugs alone [246,248]. Therefore, rivastigmine/memantine delivery by PAMAM-Lf could not only improve the drug’s brain uptake, but also translated into a clinical improvement. G4 PAMAM dendrimers were also explored as carriers of donepezil [250]. PAMAM increased the pharmacokinetic properties of donepezil and led to a higher brain concentration of donepezil after an intravenous administration in Sprague-Dawley rats, compared with pure donepezil administration. More importantly, PAMAM/donepezil nanoparticles inhibited AChE activity to a significantly higher extent than the donepezil alone when at the same molar concentration. Hence, the attachment of donepezil molecules to the dendrimers allowed the drug to interact in a stronger way with AChE, translating into a biological improvement. Lastly, G4 and G4.5 PAMAM dendrimers were used as a combinatory therapy of tacrine. Tacrine was the first AChE inhibitor licensed for AD treatment. It is the most potent and clinically effective AChE inhibitor and has good intestinal permeability due to its configuration and medium lipophilicity. However, it causes hepatotoxicity, which made its use largely discontinued [259]. Even though G4 and G4.5 PAMAM dendrimers did not interact with tacrine, their co-administration reduced tacrine’s cytotoxic effects, and its hepatoxicity in zebrafish larvae. Moreover, both G4 and G4.5 PAMAM dendrimers alone could reduce the activity of AChE, demonstrating their capacity as a drug per se. Therefore, co-administration of tacrine and PAMAM could not only decrease the toxicity of tacrine, but also have a synergetic effect on the anti-acetylcholinesterase activity.

Dendrimers have also been used as nanocarriers for compounds that alleviate amyloid pathology. Navas Guimarães et al. used a G1 PEG-GATG dendrimer as a carrier for DMHCA (N,N-dimethyl-3β-hydroxycholenamide), a liver X receptor (LXR) modulator [251]. LXR are receptors mainly produced in the CNS by astrocytes, which modulate the expression of ApoE and ATP-binding cassette transporter A1 (ABCA1) [260]. ApoE is a glycoprotein described to interact with Aβ peptide, and is one of the main contributors to Aβ drainage from the brain. Hence, LXR modulation could translate into a reduction of Aβ burden by ApoE expression. Moreover, ABCA1 controls the lipidation of ApoE, which makes the interaction of ApoE with Aβ possible [260]. Three pendant DMHCA molecules were covalently conjugated with the G1 PEG-GATG dendrimer, forming the PEG-[G1]-DMHCA. PEG-[G1]-DMHCA were successfully internalised by astrocytes and neurons in vitro and led to an enhancement in the expression of ApoE and ABCA1. Chronic intranasal administration of PEG-[G1]-DMHCA in McGill-Thy1-APP transgenic mice led to a decrease in Aβ burden and an improvement in object recognition memory compared to the groups treated with the dendrimer alone.

The decrease of Aβ burden has also been targeted by the delivery of BACE1 small interfering RNA (siRNA) [252]. In this report, the authors used a G2 6-O-α-(4-O-α-D-glucuronyl)-Dglucosyl-β-cyclodextrin(CDE)-functionalised PAMAM dendrimers as nanocarriers of a short hairpin RNA expression plasmid (shRNA) targeting BACE1 knockdown (shBACE1). By combining the G2 PAMAM dendrimers’ ability to inhibit amyloid formation and disrupt preformed fibrils with the delivery of a cargo that suppresses amyloid protein production, the authors aimed to create a multifunctional agent against amyloidosis [252]. These nanoparticles could reduce the expression of BACE1 in N2a neuronal cells, inhibit Aβ fibrillation, and disrupt preformed amyloid fibrils in vitro. More importantly, nanoparticle intravenous administration delayed the cognitive decline and reduced the brain amyloid amount in AD mice (App^NL-G-F/NL-G-F^ knock-in mouse model) compared to saline-injected AD mice. Dendrimers alone demonstrated similar in vivo properties (but to a lower extent), indicating their capacity to act as a drug per se.

G4 PAMAM-OH was also used to ameliorate tau pathology by the delivery of 2,6-Dimethoxy-4-(5-Phenyl-4-Thiophen-2-yl-1H-Imidazol-2-yl)-Phenol (DPTIP) [253]. DPTIP is an inhibitor of the neutral sphingomyelinase 2 enzyme (nSMase2), which catalyses the formation of ceramide, allowing the formation of extracellular vesicles (EVs). As a growing body of evidence suggests that tau pathology spreads between neurons by EVs, inhibiting their formation could stop the tau pathology spreading and have a neuroprotective effect in AD [261]. The chronic oral administration of DPTIP-PAMAM nanoparticles in a rapid tau propagation model (AAV-hTau) led to a robust reduction in nSMase2 activity in the tau-affected areas, which translated into a significant reduction in the propagation of tau pathology. DPTIP-PAMAM treatment resulted in lower amounts of total tau peptide and significantly fewer Tau^+^ neurons in AAV-hTau mice, compared to vehicle-treated AAV-hTau mice.

Lastly, we highlight a recent preclinical study that targeted AD-associated oxidative stress and inflammation. Zhong et al. explored angiopep-2-conjugated G3 PAMAM dendrimers as carriers of Prussian blue (PB) to improve their biodistribution and BBB permeability. PB (KFeIII [FeII(CN)_6_]) is a dark blue pigment used in the clinic for thallium poisoning [261]. More importantly, it has antioxidant properties by their scavenging capacity of ROS [261]. Therefore, it could have a neuroprotective effect on AD. These nanoparticles showed the capacity to scavenge H_2_O_2_ in vitro and reduced Aβ (1-42)-induced cytotoxicity in BV2 microglial cells. Their injection in APP/PS1 mice suppressed the activation of microglia, reduced the release of ROS, IL-1β, and TNF-α, and decreased the content of oxidation products. More importantly, PB/PAMAM nanoparticles showed to reduce amyloid plaque number, rescue neuronal function, and improve spatial learning and memory in APP/PS1 mice compared to nontreated APP/PS1 mice. Therefore, the delivery of PB by the angiopep-2-conjugated G3 PAMAM dendrimers could ameliorate AD pathology in a multifunctional manner.

## 9. The Other Side of Dendrimers—Caveats and Challenges

Despite the major advantages of dendrimers and their high potential for biomedical applications, they also possess drawbacks that should be considered.

The laborious synthesis of some of the dendritic structures can be seen as a caveat. The multi-step synthesis of dendrimers can be demanding, time-consuming, and expensive, especially for high-generation dendrimers. The higher the generation of the dendrimer is, the more reactions are necessary to synthesize it. This translates into long synthesis routes, an increased possibility of defects in the dendritic structure, and high costs. Moreover, long synthetic routes can pose implementation issues, as the final yields may be low, and the scalability of the synthesis becomes challenging. To overcome these issues, several groups have invested in accelerated methods of synthesis [262,263,264]. One example of these accelerated synthesis routes is the orthogonal coupling method. In this synthetic approach, two different branching units, which have chemoselective functional groups, are added alternately to the dendrimer/dendron. This approach eliminates the need for activation/deprotection reactions, reducing in this way the number of reactions and speeding up the synthesis [262]. One example is this synthesis route of the G4 phosphorus-containing dendrimers [265]. In this synthetic route, two branching units are used: (1) H_2_NNMeP(S)(OC_6_H_4_PPh_2_)_2_ and (2) (N_3_P(S)(OC_6_H_4_CHO)_2_) (Figure 6). By orthogonal coupling, the G4 dendrimers can be obtained in a 4-step synthesis route. More importantly, the authors showed that a one-pot approach (but multi-step) yielded a very similar product as the one obtained in a step-by-step process (with purifications at each generation). Therefore, the orthogonal coupling is especially interesting because it permits one-pot synthesis, reducing the purification steps as well.

Another accelerated method of dendrimers’ synthesis worth mentioning is click chemistry-based synthesis. Click chemistry was introduced by Sharpless et al. in 2001 [266]. It includes several highly efficient reactions with yields close to 100%. One example of the use of click chemistry for the synthesis of dendrimers is the PEGOL-60 [178]. Its synthesis is a water-based 5-step synthesis, based on Cu(I)-catalysed azide–alkyne cycloaddition (CuAAC) and thiol-ene addition reactions (Figure 7). Interestingly, the biodistribution and biological properties of PEGOL-60 in the context of inflammation were close to the ones of G4 PAMAM-OH. This suggests that the PEGOL-60 could be a powerful substitute for the PAMAM dendrimer for this application, translating into a faster and more scalable synthetic route.

Another concern in the dendrimers’ field is still the lack of understanding about the dendrimers’ effect on biochemical pathways and processes in the body. Dendrimers are macromolecules that can not only resemble biological molecules (e.g., proteins), but also interact with cellular components [70]. They can bind with the body’s vitamins, heavy metals, ions, lipids, and proteins [267]. This interaction can change the biochemical pathways in the cells and translate into cytotoxicity. In fact, several dendritic structures have demonstrated cytotoxicity towards normal and cancer cells [15,268]. Cationic dendrimers, like PAMAM, PLL and PPI dendrimers, have shown haemolytic activity, led to a reduction in erythrocyte number and an increase in leukocytes, and had the ability to induce blood clots, especially in high generations [269,270,271,272,273,274]. PAMAM dendrimers have also been reported to possess neurological toxicity. G4 PAMAM dendrimers were shown to interfere with synaptic signalling by the increase of the membrane permeability and intracellular calcium concentration by 8-fold with a complete disruption of the transients pattern in hippocampal neurons in vitro [275]. Moreover, G5 PAMAM dendrimers induced neuronal death via ROS formation and affect the proliferation and migration of human neural progenitor cells in vitro [276,277]. G5 PAMAM dendrimers have also demonstrated hepatotoxicity in vivo [278,279]. G6 PAMAM dendrimers demonstrated renal toxicity in normal and diabetic rats [280]. G3 PAMAM dendrimers promoted acute lung injury in vivo [281].

Dendrimers’ cytotoxicity was shown dependent on concentration, generation, and surface characteristics, namely on the number and nature of the terminal groups. Higher generations are associated with higher cytotoxicity than a lower generation [282,283]. G4 PAMAM dendrimers demonstrated a half maximal inhibitory concentration (IC_50_) of 3.21 µM and 1.44 µM by MTT ((3-(4,5-dimethylthazolk-2-yl)-2,5-diphenyl tetrazolium bromide) assays in human epidermal keratinocytes (HaCaT) and primary adenocarcinoma of colon (SW480) cells, respectively. On the other hand, G6 PAMAM dendrimers have an IC_50_ of 1.02 µM and 1.16 µM for the same cells. Hence, G6 PAMAM dendrimers were more cytotoxic to these cells than G4 ones. Regarding the nature of the surface moieties of dendrimers, cationic dendrimers are usually more toxic than neutral or anionic dendrimers [271,274,284]. The increased toxicity of cationic dendrimers has been suggested to be linked to their electrostatic interaction with negatively charged cell membranes. This interaction can create nanopores in the cellular membrane, leading to membrane damage, leakage of the cellular content, and ultimately cell death [15]. Cationic dendrimer toxicity has also been related to the formation of ROS, increased lysosomal activity, induction of apoptosis, and DNA damage [283]. Dendrimers’ associated cytotoxicity has been reported in PAMAM [274,282,283,285,286], PPI [274,286,287], CPD [288,289], and copper(II)-conjugated phosphorous dendrimers [290].

To improve dendrimers’ biocompatibility and therefore avoid acute cytotoxicity, a common approach is to perform surface modifications. Grafting the dendritic surface with PEG, acetyl groups, carbohydrates, and other moieties can mask the cationic charge of dendrimers and hamper their cytotoxicity [15]. For example, the grafting of maltose moieties to G4 PPI dendrimers (25%) could significantly reduce the toxicity of these dendrimers in Chinese hamster fibroblast cell line (B14), human liver hepatocellular carcinoma cell line (HepG2), mouse neuroblastoma cell line (N2a), and rat liver cell line (BRL-3A) [287].

Another limitation of the use of dendrimers and other nanostructures for biomedical applications is the lack of understanding about their impact on tissues in the longer term. Nanostructures, due to their small size, can accumulate inside cells and tissues, inducing toxicity [16]. Bioaccumulation-related toxicity can be a concerning issue, especially for filtration organs, such as the liver and kidney. A strategy to overcome this limitation is the design of biodegradable dendrimers. By the inclusion of labile bonds, such as hydrolysable bonds, within the dendrimer backbone, biodegradable dendrimers can be attained. The dendrimers’ biodegradability can not only prevent the bioaccumulation of synthetic materials in tissues, but also be a key asset in favouring the cargo release [16]. Hence, these novel nanosystems present two main advantages compared to conventional dendrimers: (1) they permit a better release of therapeutics, ideally on the target tissue, and (2) since the dendrimer is degraded into smaller parts, it can be easily drained from the body, avoiding toxicity due to synthetic material accumulation [291]. With that in mind, some research groups, including ours, have proposed new families of biodegradable dendrimers. For instance, Leiro et al. designed and developed two new families of biocompatible and biodegradable Gallic Acid–Triethylene Glycol Ester dendrimers (GATGE dendrimers)—one hybrid biodegradable [292], and another one fully biodegradable [293,294] for nanomedicine applications. Both types of dendrimers have been successfully explored as nucleic acid carriers [292,294]. Other biodegradable dendrimers used in biomedical applications are the 2,2-bis(hydroxymethyl)propanoic acid (bis-HMPA)-based [295,296], polyester [297], and PEPE dendrimers [298].

In any case, these data point to the fact that one cannot generalize the results found for one dendrimer in one application to the whole constellation of scenarios. So, biological studies need to be carefully designed and the final structure of the selected dendrimer fine-tuned for the final application.

## 10. Concluding Remarks and Future Perspectives

In the context of AD, dendritic structures can tackle several disease hallmarks by acting as drugs per se in Aβ and tau aggregation, cholinergic imbalance, inflammation, oxidative stress, and/or ionic imbalance.

From the therapeutic properties of dendrimers and derivatives discussed in this review, the most widely studied are the anti-amyloidogenic ones. Different dendritic structures have been shown to possess anti-amyloidogenic properties not only against Aβ peptide, but also against tau protein and other amyloid proteins such as α-synuclein. Since tau protein and α-synuclein aggregation have also been linked to AD [299], the anti-amyloidogenic properties of dendrimers alone could represent a multitargeted nanotherapy for AD. This property was found to depend on the dendrimer/peptide ratio, dendrimer generation, and the characteristics of the dendritic surface (chemical nature, charge, and density of the functional groups). Since Aβ peptide has hydrophobic and positively- and negatively-charged amino acid residues, distinct dendrimer/peptide interactions have been proposed based on the surface characteristics of dendrimers. Nonetheless, more densely packed surfaces with hydrophobic and/or changed groups seem to create dendrimers more prone to inhibit amyloid aggregation and attenuate its cytotoxicity.

As pointed out by Caminade et al., the backbone of the dendritic structures can have a major role in their biological properties and it is not an “innocent” feature, as it was believed in the past [185]. Even though there are several reports on the anti-amyloidogenic properties of dendrimers, to the best of our knowledge, there is only one report comparing the interaction between Aβ peptide and several dendrimers with different backbones. In this study, the authors compared the binding capacity of G5 PAMAM, G3 PPI, and G4 CPD towards Aβ (1–28) by EPR spectroscopy [300]. The results demonstrated that PAMAM dendrimers interacted more strongly with the peptides than the other dendrimers, suggesting stronger anti-amyloidogenic properties. Nevertheless, G5 PAMAM dendrimers possess a much higher multivalency (128) than G3 PPI (16) and G3 CPD (48), which suggests that these results are more likely linked to the higher multivalency of PAMAM than a result of its inner scaffold properties. Additionally, this study only compares the binding capacity of dendrimers and not their effective anti-amyloidogenic properties. Therefore, a broad study comparing the anti-amyloidogenic properties of different dendritic backbones is still an open gap in the open literature. Another issue to address in the future is the standardization of aggregation conditions in which the anti-amyloidogenic properties of dendritic structures are tested. In fact, the standardization of in vitro aggregation experiments towards reproducible and comparable data is an open discussion in the field of amyloid research [301]. Comparing the reports mentioned in this review, the conditions of aggregation vary in pH, buffer (type and molarity), and the presence or absence of aggregation facilitators, such as heparin. As these parameters modify the Aβ peptide aggregation kinetics [22,302], (quantitative) comparison between studies (and dendritic structures) performed under different aggregation conditions should be careful. This points out again the need for a comprehensive study comparing several dendritic structures in the same aggregation conditions.

Another key feature of dendritic structures is their ability to modulate inflammation. In this report, it is evidenced that dendritic structures, including dendritic polymers, can target neuroinflammation and inflammatory cells, possessing anti-inflammatory properties on their own. Their surface functionality, generation/size and internal structure were shown to influence their anti-inflammatory properties and modulate their biological behaviour. Here, higher generation and neutral (-OH) or negatively charged (-N(CH_2_P(O)(OH)(ONa))_2_ and -SO_3_^−^) dendritic structures seem more likely to exhibit anti-inflammatory properties and target neuroinflammation. In addition, an internal backbone that favours the distribution of terminal active functional groups in a packed multivalence surface seems to be crucial for their anti-inflammatory properties. In this report, both systemic inflammation and neuroinflammation were considered, as increasing evidence suggests that they can be related [154]. In addition, few studies reported the anti-inflammatory effect of dendrimers and derivatives on neuroinflammation. Future studies should focus more on the anti-inflammatory properties of dendrimers in neuroinflammation specifically and investigate if the reported systemic anti-inflammation character can be translated into an anti-inflammatory effect in the CNS and neuroglial cells. Moreover, the exploration of the anti-inflammatory capacity in other types of dendrimers could be an interesting expansion in the field.

Dendritic structures have also been described to possess inhibitory properties on AChE, and antioxidant and chelator properties. In the case of these properties, the pool of reports that target AD pathology specifically is limited and few conclusions can be drawn. Further research on these properties could be an asset for the field.

Moreover, there is a lack of in vivo studies demonstrating the biological properties of dendrimers in AD pathology. From what is known so far, there are only four in vivo studies conducted in animal models that explore the capacity of dendrimers to act as a drug per se in AD [26,118,123,252]. Since in vivo studies are key for clinical translation, analysis of the dendrimers’ effect in vivo is imperative and should be a consideration in future studies in the field.

Besides the discussed relevant biological properties to act as a drug per se, dendrimers can also be used as efficient nanocarriers of different therapeutics/drugs for AD treatment. Due to their intrinsic characteristics, such as a globular shape, well-defined structure, and high-number and highly tuneable surface functional groups, they are first-in-class systems for drug delivery [303]. Their multivalency allows them to carry multiple drug units at once or even different types of drugs, translating into a multivalent approach that can translate into a biological and clinical improvement, as aforementioned.

Due to their capacity to act simultaneously as drugs per se and as nanocarriers, dendrimers and derivatives possess a great capacity to create multifunctional nanotherapeutics for AD. Along this manuscript, we pointed out several types of dendrimers that can tackle more than one disease hallmarks at once. For example, PAMAM dendrimers have been described as anti-amyloidogenic agents, inhibitors of AChE activity, and anti-inflammatory and antioxidant agents. Additionally, they are the main nanocarrier system used for the delivery of AD-relevant drugs. Other types of dendritic structures, such as CPD, VPD, dPGS, and glycodendrimers, also possess the capacity to hamper the disease in more than one front. Nevertheless, few studies explored the multifunctionality of dendrimers in AD, with only PAMAM [252], CPD [21,24], and dPGS [131] being investigated in this framework. The multifunctionality of other dendritic structures is still unexplored, leaving room for further preclinical studies.

Even though out of the scope of this review and, therefore, not discussed here, dendrimers also possess a high potential as diagnostic agents. The proven interaction of dendrimers with amyloid species could be explored as a diagnostic tool for AD by the conjugation of dendrimers with imaging agents, like dyes, fluorophores, or contrast agents. In that way, amyloid species could be imaged in situ by techniques like magnetic resonance imaging or computerized tomography. Moreover, as dendrimers can also modulate amyloid fibrillation, their conjugation with diagnostic moieties would represent a theragnostic approach. To the best of our knowledge, no preclinical studies assessed the theragnostic capacity of dendrimer towards amyloid species, hence it is a field with room for exploration. Nevertheless, dendrimers have been explored as biosensors for the detection of AD hallmarks, such as ACh, amyloid, and tau species in biological fluids [304,305].

Finally, another issue to tackle in the future is the toxicity and degradation of dendritic structures in vivo. The implementation of dendrimers in biomedical applications depends on their safety profile. As discussed in the previous section, several reports demonstrate cytotoxicity associated with dendrimers [15]. Moreover, the proposed dendrimers for application in the context of AD lack biodegradability, which can translate into bioaccumulation and toxicity. These issues together with the lack of a comprehensive understanding about the impact that dendrimers have on biological processes (both in short- and long-time exposure) can hamper the use of dendrimers in clinical practice. Therefore, further studies must be conducted to understand the safety profile of dendrimers and more biodegradable dendritic structures should be designed and applied in the biomedical field.

In conclusion, dendrimers and dendritic structures have enormous potential as a tool to tackle multifactorial diseases like AD due to both their intrinsic properties to act as a drug per se and to function as a nanocarrier. The combination of their intrinsic properties with the therapeutical benefits of the cargo can create a multifunctional and multifaceted therapy, which can modulate a complex disease such as AD. While there is still a long road ahead, the possible beneficial impact that one can achieve with such versatile systems overrules the challenges that remain.

## Figures and Tables

**Figure 1 pharmaceutics-15-01054-f001:**
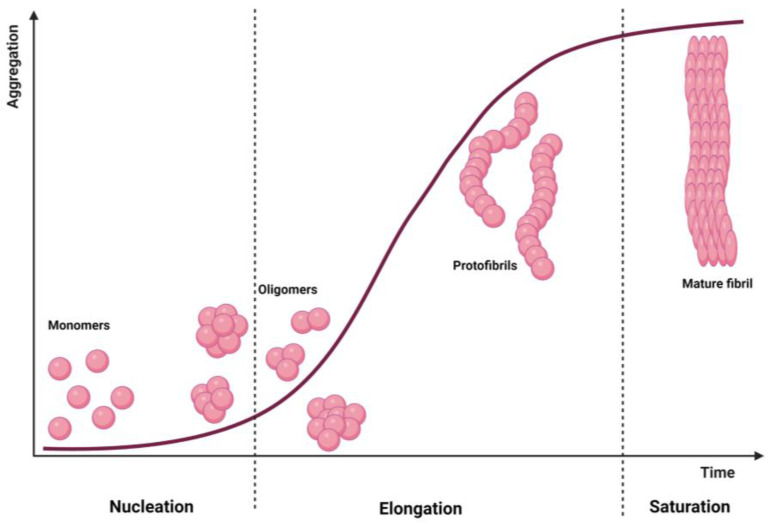
Amyloid fibril formation. The kinetic curve of a nucleation-dependent mechanism and the species expected in each moment. Created with BioRender.com.

**Figure 2 pharmaceutics-15-01054-f002:**
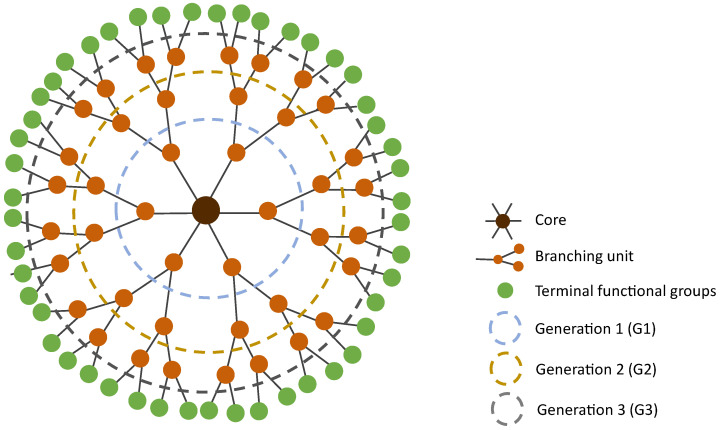
Two-dimensional representation of a spherical structure of generation 3 (G3) dendrimer.

**Figure 3 pharmaceutics-15-01054-f003:**
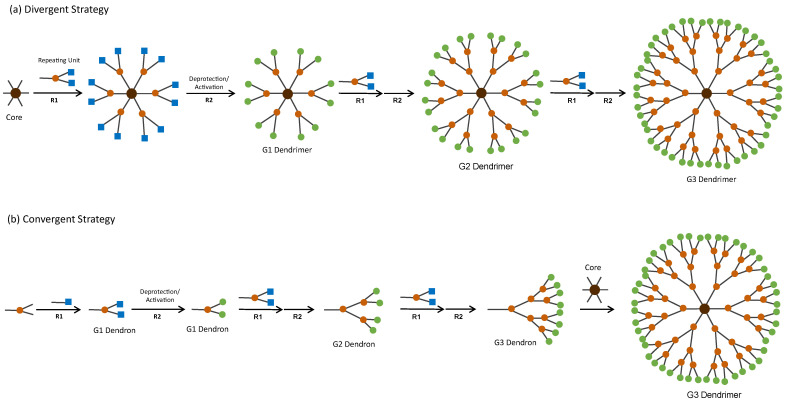
Classical strategies for the synthesis of dendrimers. Both strategies include growth reactions (R1) and deprotection/activation of the branching points (R2) to create a new generation of dendrimer/dendron. Squares represent protected/inactive functional groups and circles represent free/active functional groups. Adapted from Leiro et al. [14] © John Wiley & Sons, Inc.

**Figure 4 pharmaceutics-15-01054-f004:**
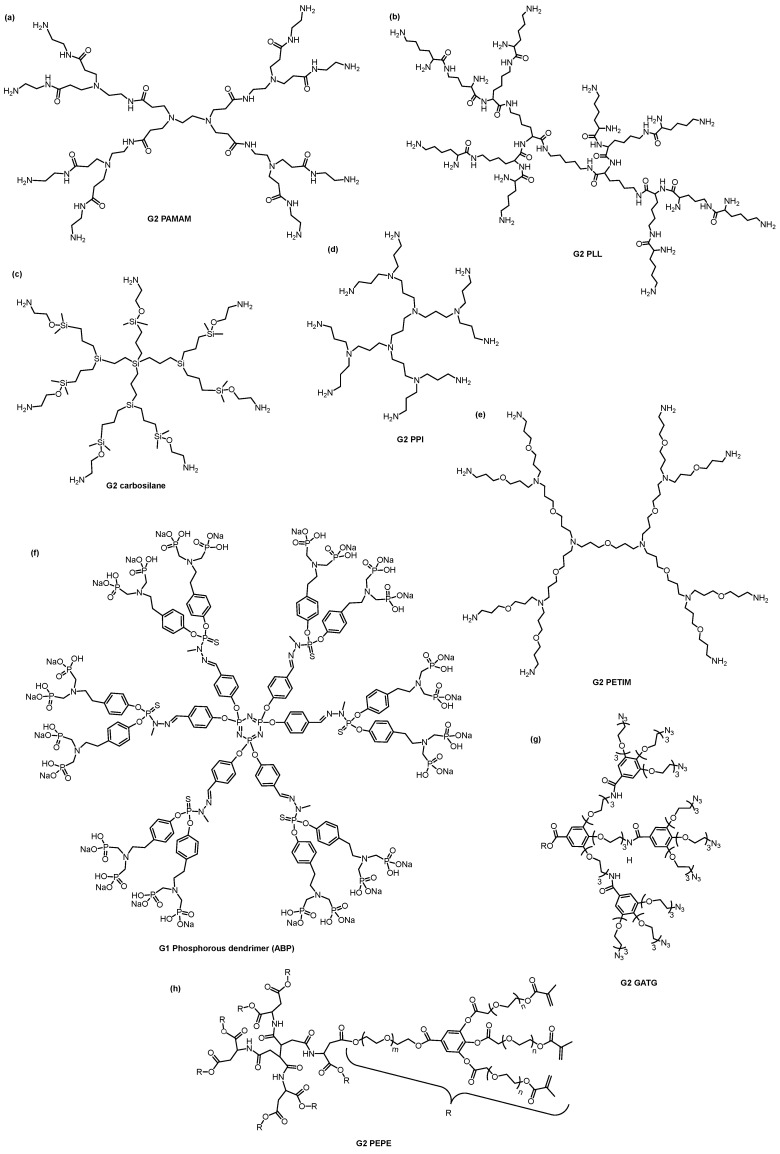
Chemical structure of the most researched dendrimers/dendrons in the biomedical field: (**a**) Generation 2 (G2) poly(amido amine) (PAMAM), (**b**) G2 poly(L-lysine) (PLL), (**c**) G2 carbosilane, (**d**) G2 poly(propylene imine) (PPI), (**e**) G2 poly(ether imine) (PETIM), (**f**) G1 azabisphosphonate-terminated (ABP) phosphorus, (**g**) G2 gallic acid-triethylene glycol (GATG) dendron, and (**h**) poly(ether)-copoly(ester) (PEPE) dendrimers.

**Figure 5 pharmaceutics-15-01054-f005:**
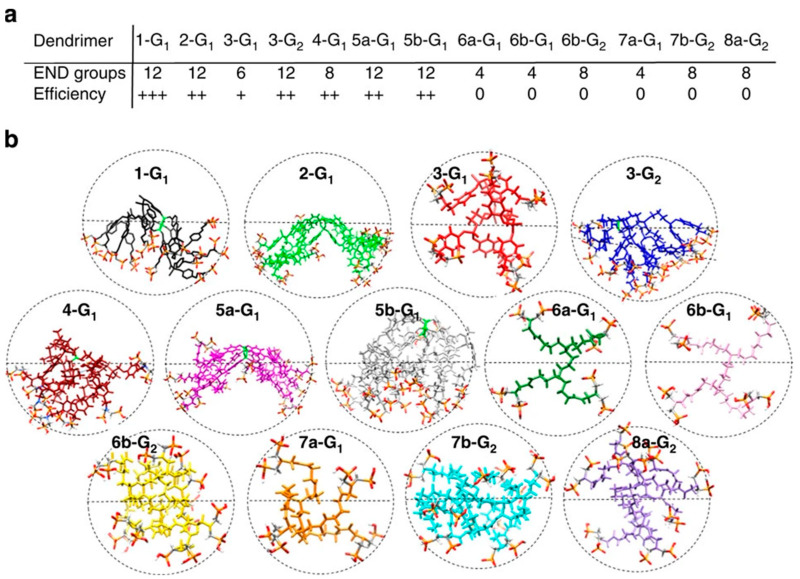
Equilibrated configurations of seven families of dendrimers—PAMAM (7a,b-Gn), PPI (6a,b-Gn), carbosilane dendrimers (4-G1), PLL (8a-G2) and phosphorus-containing dendrimers (1-G1, 2-G1,3-Gn, 5a,b-G1) bearing azabisphosphonate (ABP) groups at their surface obtained by molecular dynamics simulations. (**a**) Bioactivity of dendrimers to alternatively activate human monocytes in vitro in function of the number of ABP groups (END groups). Efficiency of activation of monocytes from 0 (no activation) to +++ (the highest activation). (**b**) Equilibrated MD configurations of different dendrimers. Adapted from Caminade et al. [185] (licensed under CC BY 4.0).

**Figure 6 pharmaceutics-15-01054-f006:**
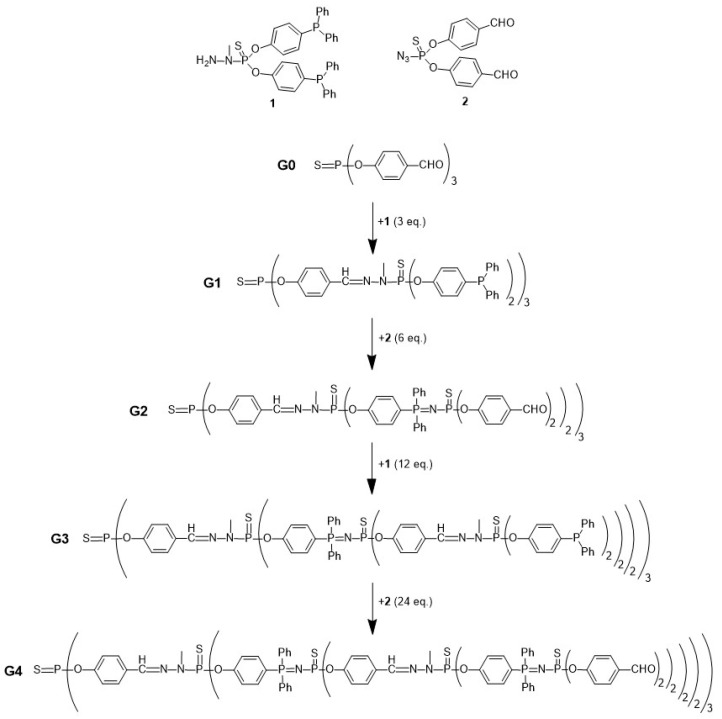
Synthesis of G4 phosphorous-containing dendrimers via the orthogonal coupling method. Synthetic route described in Brauge et al. [265].

**Figure 7 pharmaceutics-15-01054-f007:**
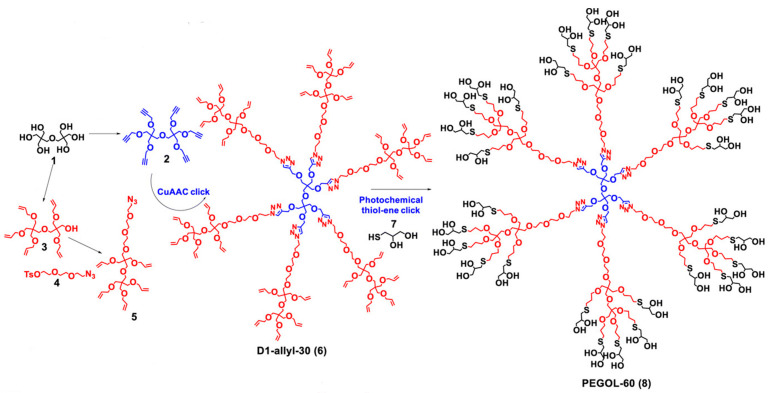
Synthetic route of polyethylene glycol–based dendrimer (PEGOL-60). Adapted from Sharma et al. [178] (licensed under CC BY 4.0).

**Table 1 pharmaceutics-15-01054-t001:** Anti-amyloidogenic properties of dendrimers and derivatives towards Aβ peptide.

Dendrimer (No. Terminal Groups (1))	Terminal Group (Charge)	Dendrimer/Peptide Ratio	Effect on Fibrillation (to Peptide Alone)	Morphology and Secondary Structure of Fibrils	Disaggregation Ability?	Attenuation of Aβ Cytotoxicity? (Cell Type)	Refs.
G3 PAMAM (32)	-NH_2_ (+)	0.0002	↑ elongation rate;	Clumps	Yes *	Yes (SH-SY5Y)	[19,104,124]
		0.002	↓ elongation rate and fibril amount (~30%)				
		0.02	↓ elongation rate and fibril amount (~ 50%)				
		0.10	Complete inhibition of aggregation				
G4 PAMAM (64)	-NH_2_ (+)	0.0002	↑ elongation rate;		Yes *	Yes (SH-SY5Y)	[104,124]
		0.002	↓ fibril amount (~30%)				
		0.02	↓ elongation rate and fibril amount (~65%)				
G5 PAMAM (128)	-NH_2_ (+)	0.0002	↑ elongation rate	Amorphous aggregates	Yes *		[104,124]
		0.002	↓ elongation rate and fibril amount (25%)				
		0.02	Complete inhibition of aggregation				
G3 CPD (48)	-NHEt (+)	0.0002	↑ elongation rate and fibril amount (~60%)	Long fibrils. Accelerated the conformational transition to β-sheet		Yes (N2a)	[21]
		0.0002	No effect	Accelerated the conformational transition to β-sheet			
		0.02–0.2	Complete inhibition of aggregation	No fibrils. Inhibition on transition to β-sheet.			
G4 CPD (96)	-NHEt (+)	0.0002	↑ elongation rate and fibril amount (~60%)	Long fibrils. Accelerated the conformational transition to β-sheet	Yes	Yes (N2a)	[21]
		0.0002	↑ elongation rate and fibril amount (~20%)				
		0.02–0.2	Complete inhibition of aggregation	No fibrils. Inhibition on transition to β-sheet.			
G3 PPI (16)	-NH_2_ (+)	0.02	↑ elongation rate; ↓ fibril amount (↓ 12%)		Yes *		[22]
		0.03	↑ elongation rate; ↓ fibril amount (↓ 56%)				
		0.04	↓ fibril amount (↓ 56%)				
		0.08	Complete inhibition of aggregation				
G4 mPPI (32 (64))	Maltose (0)	0.1	↑ elongation rate	Clumped fibrils	Yes *	Yes (PC12/SH-SY5Y)	[20,118,122]
		1	↑ elongation rate				
		5	↓ elongation rate and fibril amount (↓ ~50%)				
		10	↓ elongation rate and fibril amount (↓ ~80%)				
G4 m-IIIPPI OS (32 (64))	-NH (36%)/-NH_2_ (28%)/Maltotriose (36%) (+)	0.01	Complete inhibition of aggregation	Granular aggregates		No	[118]
G4 mPPI OS (32 (64))	-NH (37.5%)/ -NH_2_ (25%)/Maltose (37.5%) (+)	0.01	↑ nucleation rate, elongation rate and fibril amount (↑ 100%)	Fibrillar. No oligomers		Yes (SH-SY5Y)	[118]
		>0.01	Complete inhibition of aggregation				
G5 mPPI (64 (128))	-NH (10%)/Maltose (90%) (0)	0.005	↑ nucleation and elongation rate	Fibrillar. No oligomers		Yes (PC12/SH-SY5Y)	[20,118]
		0.1–10	Complete inhibition of aggregation	Amorphous aggregates			
G5 mPPI (64 (128))	Maltose (0)	0.002	↑ elongation rate and fibril amount (↑ 120%)	Acceleration of the conformational transition to β-sheet			[119]
		0.02	↑ elongation rate and fibril amount (↑ 100%)				
		0.2	Complete inhibition of aggregation	Inhibition of conformational transition to β-sheet.			
G5 m-IIIPPI (64 (128))	-NH (6%)/Maltotriose (94%) (0)	0.002	↑ elongation rate and fibril amount (↑ 100%)	Acceleration of the conformational transition to β-sheet			[119]
		0.02	↑ fibril amount (↑ ~20%)				
		0.2	Complete inhibition of aggregation	Inhibition of conformational transition to β-sheet.			
G5 mPPI SO3 (64 (128))	Maltose/ -O-SO_3_^−^ (0.76 equivalent of OH units) (−)	0.0002	↓ fibril amount (↓ ~25%)			Yes (mHippoE-18)	[120]
		0.002	↓ fibril amount (↓ ~45%)				
		0.02	↓ fibril amount (↓ ~45%)	Slows down the conformational transition to β-sheet			
G4HisMal (64 (128))	-NH (32%)/Maltose (48%)/Histine (20%) (+)	0.10	↓ elongation rate and fibril amount (↓ ~65%)	No oligomers. Mesh like fibrils		Yes (SH−SY5Y)	[26]
		1	↓ elongation rate and fibril amount (↓ ~75%)	No oligomers. More irregular aggregates			
GATG-Mor (27)	Morpholine (0)	0.0002	No effect			Yes (B14)	[110]
		0.002	↑ fibril amount (↑ 25%)				
		0.02	↑ elongation and fibril amount (↑ 116%)	Long fibrils; ↑ fibrils; Faster conformational transition to β-sheet			
2G0-GaOH (2)	Gallic acid (-OH) (0)	0.5	Complete inhibition of aggregation		Yes	Yes (SH-SY5Y)	[111]
		1	Complete inhibition of aggregation. ↓ final amount of ThT+ aggregates in cell culture (~50%).	↓ of non-fibrillar structures (~20%); ↓ Elongated fibrils. ↑ unstructured aggregates (condensed and less organized) and/or shorter fibrils			
		2	Complete inhibition of aggregation				
2G1-Ga-OH (6)	Gallic acid (-OH) (0)	0.5	Complete inhibition of aggregation		Yes	Yes (SH-SY5Y)	[111]
		1	Complete inhibition of aggregation. ↓ final amount of ThT+ aggregates in cell culture (~60%).	↓ of non-fibrillar structures (~43%); ↓ small aggregates (~10%); ↓ Elongated fibrils; ↑ unstructured aggregates (condensed and less organized) and/or shorter fibrils			
		2	Complete inhibition of aggregation				
3G1-Ga-OH (9)	Gallic acid (-OH) (0)	0.5	Complete inhibition of aggregation		Yes	Yes (SH-SY5Y)	[111]
		1	Complete inhibition of aggregation. ↓ the final amount of ThT^+^ aggregates in cell culture (~40%).	↓ of non-fibrillar structures (~32%); ↓ Elongated fibrils; ↑ unstructured aggregates (condensed and less organized) and/or shorter fibrils			
		2	Complete inhibition of aggregation				
G3 Lysine dendrimer (8)	(CH_2_)_4_NH_3_^+^(+)	0.02	↑ nucleation rate, elongation rate and fibril amount (↑ ~20%)			Yes (SH-SY5Y)	[125]
		0.1	↑ nucleation rate and elongation rate				
G5 Lysine dendrimer (26)	(CH_2_)_4_NH_3_^+^ (+)	0.02	↓ nucleation rate, elongation rate and fibril amount (↓ ~100%)				[125]
		0.1	↑ fibril amount (↑ ~50%)				
G3/G4 PAMAM-COOH (32/64)	-COOH (−)	0.1–25	No effect	Fibrillar. No change in the secondary structure of the peptides			[126]
G5 PAMAM-COOH (128)	-COOH (−)	0.1–10	No effect	Fibrillar. No change in the secondary structure of the peptides		No (SH-SY5Y)	[107]
		25	↓ fibril amount (20%)				
G6 PAMAM-COOH (256)	-COOH (−)	0.1–25	No effect	Fibrillar. No change in the secondary structure of the peptides			[126]
G3 PAMP (32)	-COOH/Phenyl groups (28.8%) (−)	0.01	No effect	Fibrillar. No change in the secondary structure of the peptides			[126]
		0.1–1	↑ elongation rate and fibril amount (up to 30%)				
G4 PAMP (64)	-COOH/ Phenyl groups (28.2%) (−)	0.01–0.5	↑ elongation rate and fibril amount (up to 30%)	Fibrillar. No change in the secondary structure of the peptides			[126]
		1	↑ elongation rate				
G5 PAMP (128)	-COOH/ Phenyl groups (7.2%) (−)	0.01	↓ fibril amount (↓ ~10%)	Fibrillar. No change in the secondary structure of the peptides		Yes (SH-SY5Y)	[107]
		0.1	↓ fibril amount (↓ ~15%)				
		0.5	↓ fibril amount (↓ ~20%)				
		1	↓ elongation rate and fibril amount (↓ ~28%)				
	-COOH/ Phenyl groups (21.2%) (−)	0.01	↓ fibril amount (↓ ~15%)	Fibrillar. No change in the secondary structure of the peptides		Yes (SH-SY5Y)	[107]
		0.1	↓ fibril amount (↓ ~20%)				
		0.5	↓ fibril amount (↓ ~30%)				
		1	↓ elongation rate and fibril amount (↓ ~38%)				
	-COOH/ Phenyl groups (30.5%) (−)	0.01	↓ fibril amount (↓ ~25%)	Irregular aggregates. Inhibition on the conformational transition to β-sheet at the equimolar ratio of peptide/dendrimer		Yes (SH-SY5Y)	[107,126]
		0.1	↓ fibril amount (↓ ~35%)				
		0.5	↓ fibril amount (↓ ~40%)				
		1	↓ elongation rate and fibril amount (↓ ~70%)				
	-COOH/ Phenyl groups (42.3%) (−)	0.01	↓ fibril amount (↓ ~25%)	Irregular aggregates. Inhibition on the conformational transition to β-sheet at the equimolar ratio of peptide/dendrimer		Yes (SH-SY5Y)	[107]
		0.1	↓ fibril amount (↓ ~40%)				
		0.5	↓ fibril amount (↓ ~50%)				
		1	↓ elongation rate and fibril amount (↓ ~70%)				
G6 PAMP (256)	-COOH/ Phenyl groups (28.8%) (−)	0.01–1	↓ elongation rate and fibril amount in a concentration-dependent manner. At peptide/dendrimer ratio 1:0.5 ↓ fibril amount by 70%	Irregular aggregates. Inhibition on the conformational transition to β-sheet at the equimolar ratio of peptide/dendrimer			[126]
G5 PAMP-OH (128)	-OH/Phenyl groups (28.8%) (+)	0.1–1	No effect	No effect at different peptide/dendrimer ratios			[107]
APD (8)	-SO_3_^−^ (50%)/CH_2_CH_2_CH_3_ (50%) (−)	0.125	↓ elongation rate and fibril amount to 66.5%	Fibrillar structures	Yes		[127]
		0.2	↓ elongation rate and fibril amount to ~60%	Fibrillar structures			
		1	↓ elongation rate and fibril amount to ~10%	Amorphous aggregates and fibrils		Yes (primary murine neuronal cells)	
		0.75	↓ elongation rate and fibril amount to 52%	Amorphous aggregates and few fibrils			
		4	Complete inhibition of fibrillation	Amorphous aggregates; Inhibition of transition to β-sheet		Yes (primary murine neuronal cells)	
SA-D (32)	-SO_3_^−^ (50%)/CH_2_CH_2_CH_3_ (50%) (−)	0.125	↓ elongation rate and fibril amount to 66.3%	Amorphous aggregates and fibrils	Yes		[127]
		0.333333333	↓ elongation rate and fibril amount to 50.4%	Amorphous aggregates and few fibrils			
		0.5–1	Complete inhibition of fibrillation	Amorphous aggregates		Yes (primary murine neuronal cells)	

(1) = number of possible substituents; * Disaggregation ability towards Prion Protein; ↑ increase; ↓ decrease; 2G0-GaOH = G0 GATG dendrimer functionalised with gallic acid groups on a bifunctional core; 2G1-Ga-OH = G1 GATG functionalised with gallic acid groups on a bifunctional core; 3G1-Ga-OH = G1 GATG dendrimer functionalised with gallic acid groups on a trifunctional core; APD = amphiphilic polyphenylene dendrons functionalised with sulfonic acid and n-propyl groups; PD = cationic phosphorous dendrimer; G4HisMal = poly(propylene imine) dendrimers with a histidine-maltose shell; GATG-Mor = gallic acid-triethylene glycol functionalised with morpholine groups; m-IIIPPI = poly(prolylene imine)-Maltotriose; m-IIIPPI OS = poly(prolylene imine)-Maltotriose open shell; mPPI = poly(prolylene imine)-Maltose; mPPI-SO3 = poly(prolylene imine)-Maltose modified with 1–2 sulphate units per maltose unit in the outer shell; PAMAM = poly(amido amine); PAMAM-COOH = carboxyl-terminated poly(amido amine); PAMP = phenyl-derivatized carboxyl-terminated poly(amido amine); PAMP-OH = phenyl-derivatized hydroxyl-terminated poly(amido amine); PPI = poly(prolylene imine); SA-D = four APD biotin-terminated assembled onto the protein streptavidin; VPD = Viologen-Phosphorus Dendrimers.

**Table 2 pharmaceutics-15-01054-t002:** Anti-inflammatory properties and uptake of dendritic structures by inflammatory cells.

Dendrimer (No. Terminal Groups)	Terminal Group (Charge)	Type of Assay	Biodistribution & Cellular Uptake	Effect on Inflammation	Refs.
G4 PAMAM (64)	-NH_2_ (+)	in vivo/ in vitro	*Biodistribution: CP newborn rabbit model* • i.p.: only found at the injection site. • i.v.: only present inside blood vessels *Cellular uptake: BV2 microglial cells* • Dendrimer uptake: 0.093 ± 0.01 pg/cell (resting state) & 0.517 ± 0.01 pg/cell (LPS-activated) • 2-fold lower uptake than neutral G4 PAMAM-OH	In vivo *Carrageenan-induced paw edema in Rats* • 35.50 ± 1.64% inhibitory activity of paw edema at 16 mg/kg dose, 4 h post-administration Cotton Pellet Test in Rats • Exhibited higher inflammatory inhibition than indomethacin alone *Adjuvant-induced Arthritis in Rats* • Exhibited higher inflammatory activity than indomethacin alone (25 ± 2.1 vs. 18 ± 0.7, *p* < 0.05) In vitro • Exhibited COX-1 and COX-2 inhibition and ~50% reduction of the NO production on LPS-activated rat peritoneal macrophages	[160,175,176]
	-OH (0)	in vivo/ in vitro	*Biodistribution: CP newborn rabbit model* • i.p.: found within the cells several millimeters. • i.v.: Co-localization with activated microglia in CP kits (but not in healthy age-matched) 4 h post-injection. Extent of dendrimer uptake correlated with the extent of disease *Cellular uptake: BV2 microglial cells* • Maximal Dendrimer uptake (resting): 0.201 ± 0.02 (pg/cell) • 4.05-fold increase in maximal dendrimer uptake by LPS stimulation (0.814 ± 0.08 pg/cell)	In vivo *Carrageenan-induced paw edema in Rats* • 31.22 ± 1.58% inhibitory activity towards paw edema at 16 mg/kg dose, 4 h post-administration In vitro • COX-2 inhibition (not COX-1) and ~50% reduction of NO production on LPS-activated rat peritoneal macrophages	[83,160,175,176,177]
G3.5 PAMAM (64)	-COOH (−)	in vivo	*Biodistribution: CP newborn rabbit model (i.v.)* • i.v.: co-localized with microglial cells 24h post-injection *Cellular uptake: BV2 microglial cells* • Maximal Dendrimer uptake (resting): 0.234 ± 0.01 pg/cell • 1.72-fold increase in maximal dendrimer uptake by LPS stimulation (0.404 ± 0.01 pg/cell)	In vivo *Carrageenan-induced paw edema in Rats* • 11.00 ± 1.60% inhibitory activity towards paw edema at 16 mg/kg dose, 4 h post-administration In vitro • ~43% reduction of the NO production on LPS-activated rat peritoneal macrophages. Inhibitory activity was significantly lower than G4 PAMAM-NH2 and G4 PAMAM-OH (*p* < 0.05) • Did not exhibit COX inhibition on LPS-activated rat peritoneal macrophages	[160,175,176]
	-COOH/Glucosamine (14%) (−)	in vitro		in vitro • Significant reduction in the release of the pro-inflammatory chemokines (MIP-1α, MIP-1β, IL-8, TNF-α, IL-1β and IL-6) in LPS-activated human PDMC • Significant reduction in the release of the pro-inflammatory chemokines (MIP-1β and TNF-α in LPS-activated human DC and MDM • Inhibited lymphocyte proliferation in the mixed leukocyte reaction and prevent INF-γ production	[159]
G6 PAMAM (256)	-OH (0)	in vivo/ in vitro	*Biodistribution: CP newborn rabbit model (i.v.)* • Increased circulation time comparing to G4 • Brain uptake higher than G4 • Extent of dendrimer uptake correlated with the extent of disease *Cellular uptake: BV2 microglial cells* • Maximal Dendrimer uptake (resting): 0.038 ± 0.03 pg/cell • 6.25-fold increase in maximal dendrimer uptake by LPS stimulation (0.377 ± 0.01 pg/cell)		[175,176]
G3/G4 CCPD (48/96)	-NHEt (+)	in vitro		In vitro • Decreased TNF-α release in LPS-activated BV-2 cells, to the same extent as NAC. • No differences in the anti-inflammatory properties between G3 and G4 • Scavenger capacity of DPPH free radicals • Exhibit reduction capacity of ferric ions. G4 had a stronger reduction capacity than G3	[24]
PEGOL-60 (60)	-OH (0)	in vivo/ in vitro	*Biodistribution: CP newborn rabbit model (i.v.)* • 10-fold higher brain uptake than age-matched healthy kits • Co-localization with activated microglia within 1h post-administration • No co-localization with activated microglia in healthy controls	In vitro • Co-treatment with LPS led to a significant downregulation in TNF-α, IL-6, IL-10, and iNOS, and upregulation in CD206, Arg1, IL-4 • Significant reduction in excreted TNF-α and NO in LPS-activated BV2 microglial • Pre-treatment resulted in a significant improvement in cellular viability upon a H_2_O_2_ challenge in BV2 microglial cells.	[178]
G0/G1 “Click dendrimers” (3 or 6)	-OH (0)	in vitro		In vitro • Reduction of NO release in LPS-activated N9 microglial cells • Inhibition of PGE2 in LPS-activated N9 microglial cells • Higher anti-inflammatory effect of G1 than G0	[179]
	-Acetylene (0)	in vitro		In vitro • Reduction of NO release in LPS-activated N9 microglial cells only at the highest concentration (50 µM) • Inhibition of PGE2 in LPS-activated N9 microglial cells • Lower anti-inflammatory effect of G1 than G0	[179]
dPG (64)	-OH (0)	in vivo/ in vitro	*Cellular uptake* • Increased cellular uptake with increased size in Macrophage-differentiated THP-1 cells • Lower particle uptake than negative dPGS for similar sized particles in Macrophage-differentiated THP-1 cells	in vivo *TLR2-/GFAP-luciferase transgenic mice—i.n. with LPS* • dPG could not suppress the LPS-activated effects on microglia and astrocytes *Mouse Organotypic Hippocampal Brain Slices* • Did not prevent spine loss caused by an Aβ insult in vitro • Three-hour pre-treatment did not change the mitochondrial activity, NO nor the cytokine release in LPS-activated N9 neonatal murine microglia cells	[131,164,180,181]
	-OSO_3_^−^ (−)	in vivo/ in vitro	*Cellular uptake* • Selectively taken up by microglial cells and not astrocytes or neurons in Mouse Organotypic Hippocampal Brain Slices • Higher uptake by microglia than astrocytes in mixed mouse cortical cultures • Higher particle uptake than neutral dPG for similar sized particles in macrophage-differentiated THP-1 cells • Increased cellular uptake with increased size in macrophage-differentiated THP-1 cells	in vivo *TLR2-/GFAP-luciferase transgenic mice—i.n. with LPS* • Reduction of microglial activation (but not astrocyte activation) in a concentration and time-dependent manner *Mouse model of complement activation* • Pre-treatment reduced C5a generation in a LPS-stimulated mice *Mouse contact dermatitis model* • Reduced leukocyte extravasation to inflamed tissue in vitro • Act as a scavenger for IL-6 and LCN2 • Directly bind to L-selectin, P-selectin, complement factor C3 and C5 • Three-hour pre-treatment inhibited the reduction of mitochondrial activity of LPS-activated N9 cells in ~25% • Three-hour pre-treatment decreased the release of NO, TNF-α and IL-6, and reduced the nitrosylated proteins in LPS-activated N9cells • Decreased the amount of Aβ internalized by neuroglia in mixed mouse cortical cultures *Mouse Organotypic Hippocampal Brain Slices* • Three-hour pre-treatment decreased the release of NO, TNF-α and IL-6 in LPS-activated slices • Three-hour pre-treatment prevented spine loss caused by LPS stimulation. • Avoided the morphological changes on postsynaptic dendritic spines of an Aβ (1-42) insult. • Reduced LCN2 production in Aβ-exposed slice, comparing to Aβ (1-42) incubation alone	[131,163,164,180,181]
G1 PPH (10/12)	-N(CH_2_P(O)(OH)(ONa))_2_ (ABP) (−)	in vivo/ in vitro		In vivo *Mouse arthritis model (IL-1-ra−/−)—i.v.* • Completely inhibited inflammation and arthritis at doses of 1 and 10 mg/kg • Decreased serum concentrations of IL-1β, TNF-α, IL-6, and IL-17 and the amount of MMP3 and MMP9 *K/BxN serum transfer mouse model—i.v.* • Prevented inflammation and arthritis In vitro • Activate human monocytes in an alternative pathway • Human monocytes remained viable longer than control monocytes, underwent phenotypical changes and increased NF-kB. • Skew mice splenocytes towards an anti-inflammatory phenotype, as it increased their production of anti-inflammatory cytokines (IL-4 and IL-10) and reduced the production of pro-inflammatory cytokines (IL-1β, TNF-α, IL-6, IL-17, IL-2, INF-γ)	[165,169,182,183,184]
	-CH(NHMe)(P(O)(OH)(ONa)) (AMP) (−)	in vivo		In vivo *Mouse arthritis model (IL-1-ra*^−/−^*)—i.v.* • Did not change paw swelling and in arthritis score	[169]
	-COONa	in vitro		In vitro • Lower activation of human monocytes than dendrimers ending with phosphonic acid groups	[182]
G1 PPH (2,4,6,8,16)	-N(CH_2_P(O)(OH)(ONa))_2_ (ABP) (-)	in vivo/ in vitro		In vivo *K/BxN serum transfer mouse model—i.v.* • Could not prevent inflammation and arthritis. In vitro • G1 PPH (2,4,6): Lower activity towards human monocytes than ABP-12 • G1 PPH (8): Most bioactive dendrimer to activate human monocytes • G1 PPH (16): Similar activity towards human monocytes to ABP-12	[183,184]
G3/G4 PPH (48/96)	-N(CH_2_P(O)(OCH_3_)_2_)_2_ (0)			In vivo *Mouse air pouch injected with zymosan—i.v.* • Reduced the number of migrating cells into the pouch • Decreased NO levels and reduced the iNOS and CD86 expression in infiltrating cells and cells lining the air pouch cavity. CD163 expression was restored in these cells. • Able to modulate M1/M2 ratio in vivo In vitro • Reduced NO, TNF-α and IL-1β release and prevent IL-4 decrease in LPS-activated mouse peritoneal macrophages. G4 > G3. • Reduced iNOS and CD86 expression in LPS-activated mouse peritoneal macrophages G4 > G3 • Prevent macrophages polarization to M1 state and return the M1/M2 balance. • Similar anti-inflammatory properties on monocyte-derived human macrophages	[170]
G1 Carbosilane (8)	-N(CH_2_P(O)(OH)(ONa))_2_ (ABP) (−)	in vitro		In vitro • Good activity in alternatively-activation of human monocytes	[185]
G2 PPI (8)	-N(CH_2_P(O)(OH)(ONa))_2_ (ABP) (−)	in vivo/ in vitro		In vivo *Mouse arthritis model (IL-1-ra*^−/−^*)—i.v.* • Did not change paw swelling and in arthritis score In vitro • Did not activate human monocytes	[169,185]
G1/G2 PAMAM (4/8)	-N(CH_2_P(O)(OH)(ONa))_2_ (ABP) (−)	in vitro		In vitro • Did not activate human monocytes	[185]
G2 PLL (8)	-N(CH_2_P(O)(OH)(ONa))_2_ (ABP) (−)	in vitro		In vitro • Did not activate human monocytes	[185]

CPD = cationic phosphorous-based dendrimer; DPPH = 1,1-diphenyl-2-picrylhydrazyl; G = generation; HI = hypoxic-ischemic; i.n. = intranasal administration; i.p. = Intraparenchymal injection; i.v. = intravenous administration; IL = interleukin; INF = interferon; iNOS = inducible nitric oxide synthase; LCN2 = lipocalin-2; LPS = Lipopolysaccharide toxin; MDM = monocyte-derived macrophages; MIP = macrophage inflammatory protein; NAC = N-acetyl-L-cysteine; PAMAM = Polyamidoamine dendrimers; PBMC = peripheral blood mononuclear cells; PEGOL = polyethylene glycol–based dendrimer; PPH = Polyphosphorhydrazone dendrimers; PLL = Poly(L-Lysine) dendrimer; PPI = Polypropyleneimine; TNF = tumor necrosis factor.

## Data Availability

Not applicable.

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
