# Peer review of "Dendrimers and Derivatives as Multifunctional Nanotherapeutics for Alzheimer’s Disease"

_pharmaceutics, 2023, doi:10.3390/pharmaceutics15041054_

Round 1

Reviewer 1 Report

The paper by A.P. Pego et al is an excellent review about the potential use of dendrimers against Alzheimer’s disease. It is well organized, with an abundant bibliography (250 references), and an interesting conclusion and perspectives. I have only minor remarks.

-          As many abbreviations are used all along the manuscript, it would be helpful to have a list of abbreviations

-          There are very few Figures in this manuscript. I would be nice to have the chemical structure of all dendrimers cited, with the abbreviation of their name, which is used all along the manuscript.

-          Figure 2 is not the best way to represent a dendrimer. At first glance, it looks like an hyperbranched polymer, because the perspective is difficult to see. At least is should be indicated that it is the 3D structure. And technically, it is more a G2 than a G3 (for G3, there should be two small branches emanating from each terminal “bowl”).

-          In Figure 3, it seems that there is a single step to grow one generation, which is most generally false. I at least 90% of the synthetic methods, two steps are needed to grow one generation. This Figure should be modified.

-          The poly(phosphorhydrazone) dendrimers are generally named PPH. Why the authors named them PHH all along the paper?

Very few typos or grammar errors:

-          line 309: “the ability to carrier therapeutics” should be better “the ability to carry therapeutics”

-          line 535: “a densely packed surface that allow” should be “a densely packed surface that allows” (s at allow)

-          line 579: “changed groups” should be “charged groups”

-          lines 620-621: something is missing or wrong in this sentence. May be change “their effect has different,” to “their effect is different,”?

-          “dendrimer 1” (line 636) and dendrimer 2 (line 641) should be defined

-          lines 908-910: in ref 156, the number of terminal ABP functions is not “2, 4, 8, 10, 12, 16 or 20” but “2, 4, 6, 8, 10, 12, 16 or 30”.

-          The sentence lines 928-930 “8-ABP inactivity in vivo was related to the dendrimer degradation through the hydrazone group, which results in an inactive counterpart” needs to have a reference

-          Line 939 “dendrimers could alternatively activate monomers”. I suppose that it should be “dendrimers could alternatively activate monocytes”

-          Line 942: “Active dendrimers were overall more hydrophilic”. No, it is exactly the contrary, at least in the context in which is this sentence.

Author Response

We thank all the comments and valuable suggestions kindly made by the reviewer #1. We answer in detail below (written in green) to the reviewer #1 questions and suggestions (in black). As referred to in our answers, we have introduced several modifications in the manuscript and, consequently, we hope that our report is accepted for publication in Pharmaceutics.

  1. As many abbreviations are used all along the manuscript, it would be helpful to have a list of abbreviations

We thank Reviewer #1 for the suggestion. We added a list of abbreviations of the most relevant terms at the end of the manuscript (lines 1548-1565).

  1. There are very few Figures in this manuscript. I would be nice to have the chemical structure of all dendrimers cited, with the abbreviation of their name, which is used all along the manuscript.

We thank Reviewer #1 for the suggestion of an additional figure. We have now included a new figure (Figure 4, section 3, lines 310-315) in which we depict the chemical structure of the most researched/applied dendrimers in the biomedical field.

  1. Figure 2 is not the best way to represent a dendrimer. At first glance, it looks like an hyperbranched polymer, because the perspective is difficult to see. At least is should be indicated that it is the 3D structure. And technically, it is more a G2 than a G3 (for G3, there should be two small branches emanating from each terminal “bowl”).

Thank you for your comment. We have now introduced a new figure that we believe can better represent a dendrimer features in a more straightforward way.

  1. In Figure 3, it seems that there is a single step to grow one generation, which is most generally false. I at least 90% of the synthetic methods, two steps are needed to grow one generation. This Figure should be modified.

Thank you for your suggestion. We have now present a modified version of the figure that further illustrates the deprotection/activation steps and growth reactions needed to grow the dendrimer in some synthetic routes.

  1. The poly(phosphorhydrazone) dendrimers are generally named PPH. Why the authors named them PHH all along the paper?

We thank Reviewer #1 for drawing our attention to this mistake. We have now corrected the abbreviation throughout the manuscript.

Typos or grammar errors:

We thank Reviewer #1 for drawing our attention to some typos and grammar errors identified in the submitted manuscript, which have now been corrected.

Reviewer 2 Report

This is a very nice review. Very informative. Authors did a great job by collecting almost all existing knowledge in the field. The list of 250 references is impressive. The tables are very useful for a quick look. Nice figures. All together - this is a review worth publishing

Author Response

We thank the encouraging comments and positive feedback given by Reviewer #2.

Reviewer 3 Report

Comments: Few critical points should clarify before being accepted.

1-    Dendrimers reached clinical trials were not described. It will be important if the authors summarized if there are clinical trials done for dendrimers and what are their identifiers.

2-    Authors did not study the disadvantages of dendrimers and their toxicity on other human cells

3-    From the viewing point of the authors, why they considered dendrimers in the treatment of  Alzheimer’s Disease? What is the advantage of dendrimers compared to other nanoparticles?

4-    Refs should be arranged according to a style of pharmaceutics MDPI Journal.

5-    How dendrimers can overcome immune system recognition, was not pointed out.

Author Response

We thank all the comments and valuable suggestions kindly made by the reviewer #3 (presented in black) that we answer in detail below (written in green). As referred to in our answers, we have introduced several modifications in this revised version of the manuscript and, consequently, we hope that our report is accepted for publication in Pharmaceutics.

  1. Dendrimers reached clinical trials were not described. It will be important if the authors summarized if there are clinical trials done for dendrimers and what are their identifiers.

We thank Reviewer #3 for this suggestion. We agree that the discussion of the dendrimers currently in clinical trials is pertinent. Hence, we have included an additional paragraph with this information in the section “3. Dendrimers – A Multivalent and Multifunctional Nanocarrier” (pages 10-11, lines 330-373).

  1. Authors did not study the disadvantages of dendrimers and their toxicity on other human cells.

We thank Reviewer #3 for this suggestion. We agree that the discussion of the disadvantages and limitations of the use dendrimers is relevant and offer a more pertinent and critical perspective to the readers. Therefore, a new section was added in pages 38-41. This is entitled “9. The other side of dendrimers – Caveats and Challenges” and covers the caveats of some dendritic structures and their toxicity.

  1. From the viewing point of the authors, why they considered dendrimers in the treatment of Alzheimer’s Disease? What is the advantage of dendrimers compared to other nanoparticles?

Thank you for your question and comment. As discussed along the previous manuscript, considering the key features of dendrimers compared to other nanostructures, we believe dendrimers are key players in the future of biomedical fields. Taking this, together with the fact that several dendritic structures can have a therapeutical effect on their own, dendrimers become powerful tools for the tackling of several diseases (and not only Alzheimer’s disease (AD)). In the case of AD, they are specially interesting because they show anti-amyloidogenic properties, which can tackle one of major causes of the disease. Moreover, they possess therapeutical properties in other disease hallmarks, such as oxidative stress and inflammation, making them powerful multifunctional therapeutics for AD. To make these points more evident, we added a paragraph in page 7-8 of the section 3 (lines 272-298), stating the advantages of dendrimers.  We also included the sentence "As these properties could translate into a clinical improvement in the context of AD, dendrimers pose as powerful tool for the treatment of AD" (lines 377-379) to correlate its intrinsic properties with its therapeutic properties.

  1. Refs should be arranged according to a style of pharmaceutics MDPI Journal.

The reference style was updated to the style of the pharmaceutics MDPI Journal.

  1. How dendrimers can overcome immune system recognition, was not pointed out.

We thank reviewer #3 from this comment. Indeed, this a relevant issue that was not sufficiently addressed in the first version of the manuscript. This discussion was now included. The following information has been added to section 6 (page 34, lines 1086-1097), of the revised manuscript.

Round 2

Reviewer 3 Report

Review manuscript is improved and can be accepted